# REASONING-TO-ENCODER DISTILLATION FOR RECOMMENDATION

## ABSTRACT

Large Language Models (LLMs) have significantly advanced recommendation systems by leveraging their extensive knowledge and reasoning skills. However, applying them to large-scale systems faces two main problems: prohibitive inference latency, especially in autoregressive models, and the generation of misaligned reasoning that is not grounded in actual user preferences. Existing distillation methods attempt to solve these problems but often fall short either by failing to transfer the essential reasoning capabilities of LLMs or by distilling flawed, misaligned reasoning, which compromises the performance and reliability of the student model. To address these challenges, we introduce a new framework, **Reasoning-to-Encoder Distillation (R2END)**. This framework is designed to effectively transfer an LLM's complex reasoning into an efficient, embedding-based architecture. To ensure the distilled reasoning is grounded in actual user behavior, we employ an "oracle-guided" process where the ground-truth item is provided to the LLM to generate a well-aligned reasoning. This reasoning is then distilled into a text encoder, which learns to create a "reasoning-infused" embedding from user history, eliminating the need for the LLM during inference. Extensive experiments on three benchmark datasets demonstrate that our method substantially outperforms state-of-the-art distillation-based methods in terms of both accuracy and diversity of recommendations. Most importantly, R2END drastically reduces inference latency and computational costs, demonstrating that it provides a practical and efficient approach to creating scalable recommendation systems that benefit from the deep reasoning capabilities of LLMs.

## 1 INTRODUCTION

Large language models (LLMs) have emerged as a powerful paradigm for enhancing modern recommender systems (Bao et al., 2023; Yuan et al., 2023; Kim et al., 2025; Sheng et al., 2025). By leveraging their extensive world knowledge and sophisticated reasoning abilities, LLMs can move beyond traditional collaborative filtering (CF) to understand the nuanced, causal relationships behind user preferences. This has led to significant improvements in recommendation accuracy, diversity, and explainability (Liu et al., 2025; Chen et al., 2025; Han et al., 2025). Researchers have explored various approaches, from fine-tuning LLMs as end-to-end recommenders to integrating them as components within existing frameworks, all aiming to harness their deep contextual understanding for more intelligent recommendations (Wu et al., 2024; Zhao et al., 2024).

Leveraging the reasoning capabilities of LLMs has become a central focus of recent research (Hüyük et al., 2025; Luo et al., 2025). To make these powerful, yet computationally expensive, abilities practical for real-world applications, knowledge distillation has emerged as a prominent technique for transferring them to smaller, more efficient models (Gu et al., 2024; Panigrahi et al., 2025). This trend is also being attempted in the recommendation domain, where various studies are exploring methods to distill the nuanced reasoning of LLMs, aiming to build recommender systems that are both intelligent and scalable (Wang et al., 2024a;b).

Despite their promise, existing methods for integrating LLMs into recommender systems face a critical trade-off between reasoning depth and practical efficiency. On one hand, using autoregressive generations directly for inference provides rich reasoning but incurs prohibitive latency and computational costs, rendering it unsuitable for large-scale, real-time applications. On the other hand,

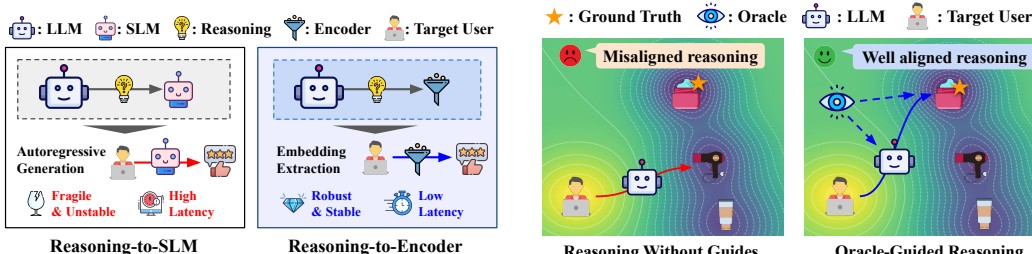

(a) Comparison of student model architectures for reasoning distillation.

(b) Comparison between unguided reasoning and oracle-guided reasoning.

Figure 1: Core motivations of our reasoning distillation framework. Text encoders are computationally efficient and enable effective learning even with limited dataset. Since the Teacher LLM may generate misaligned reasoning, it is necessary to generate appropriate reasoning through oracle guidance.

distillation methods that attempt to transfer LLM's reasoning abilities to small language models (SLMs) often fall short in recommendations (Wang et al., 2024a;b). We observe that these methods typically distill flawed or misaligned reasoning, as the teacher LLM frequently generates misaligned reasoning when recommending items without knowing the ground truth. This critical flaw means the student model learns from incorrect rationales, fundamentally limiting its performance and reliability.

To address the aforementioned challenges, we propose **R**easoning-**to**-**En**coder **D**istillation for Recommendation (**R2END**), a novel framework designed to transfer aligned reasoning to an efficient encoder-based architecture. Our core design choice is to distill an LLM's reasoning capabilities into a lightweight text encoder, which learns to produce a **reasoning-infused** user embedding that captures the LLM's rationale without its inference overhead. To ensure the reasoning signal for this distillation is aligned with user behavior, we employ an **oracle-guided** process to generate it from a teacher LLM.

**Reasoning-to-Encoder Distillation.** We distill the reasoning from teacher model not into an SLM, but into a computationally efficient text encoder. This student encoder learns to produce a reasoning-infused embedding directly from a user's history, capturing the LLM's reasoning process in a compact, fast-to-compute vector representation. As illustrated in Figure 1a, our choice to distill reasoning into a text encoder offers a dual advantage over fine-tuning a generative SLM: significant improvement in computational efficiency, and a superior learning effectiveness, particularly in data-limited settings. We attribute this effectiveness to two primary factors. From a task complexity perspective, training an encoder to predict a single semantic embedding is a far more constrained and sample-efficient objective than training an SLM for high-dimensional, token-by-token generation. Furthermore, the target signal is more robust; a pretrained encoder maps semantically similar, yet syntactically different, rationales to close points in the embedding space, providing a consistent learning target. In contrast, the token-level objectives of supervised fine tuning (SFT) are sensitive to superficial variations in the teacher's output, making the learning process less stable.

**Oracle-guided Reasoning Generation.** We introduce an oracle-guided generation process where the ground-truth item is provided to the LLM, compelling it to produce high-fidelity reasoning that is aligned with user behavior. Our experiments revealed that even when employing an LLM with over 10 billion parameters, the proportion of instances where the ground-truth item is ranked within the top-10 was surprisingly low, falling below 7%. This finding underscores a fundamental risk: naively distilling the LLM's raw output would inevitably force a student model to learn from flawed and misaligned rationales. Furthermore, while common strategies like rejection sampling can filter these incorrect instances, they drastically reduce the volume of viable training data, thereby hindering the effective distillation of the LLM's reasoning capabilities. This motivates our oracle-guided generation process. As illustrated in Figure 1b, by providing the ground-truth item to the LLM as an "oracle," we compel it to generate reasoning that is explicitly aligned with the user's actual behavior. This ensures the creation of a high-fidelity and trustworthy knowledge source, paving the way for a more effective and reliable distillation process.

Through extensive experiments on three real-world datasets, we demonstrate the effectiveness of our proposed method. R2END significantly outperforms state-of-the-art distillation-based recommenders in terms of accuracy and diversity. Crucially, by eliminating the need for an LLM at inference time, our approach drastically reduces latency and computational costs by orders of magnitude. These results validate that R2END offers a practical and effective pathway to building scalable recommendation systems that successfully incorporate the reasoning capabilities of LLMs.

**Contributions.** Our main contributions are as follows. First, we tackle the limitations of current distillation methods that rely on SLMs, which suffer from inference inefficiency and training instabilities. In addition, we empirically identify and analyze the critical issue of reasoning misalignment in LLMs when applied to recommendation tasks, demonstrating that unguided reasoning is often unreliable. Second, we propose a novel and practical framework, Reasoning-to-Encoder Distillation for Recommendation, which effectively addresses this issue by aligning the LLM's rationale with ground-truth user behavior before distilling it into an efficient text encoder. Third, through extensive experiments, we validate that R2END not only achieves state-of-the-art performance among distillation-based methods, but also drastically reduces inference latency, proving its viability for real-world, large-scale deployment.

## 2 RELATED WORK

### 2.1 LLM-BASED RECOMMENDATION

The integration of LLMs into recommender systems has opened new frontiers. Research in this area is primarily branching into two main approaches: generative methods and embedding-based methods. Recent studies have further demonstrated that LLMs can enhance recommendation diversity, mitigate the cold-start problem (Kim et al., 2024; Liu et al., 2025; 2024), and provide greater explainability (Ramos et al., 2024), leading to a wide range of research aimed at leveraging these distinct advantages.

Early approaches leveraged the autoregressive capabilities of LLMs to directly generate recommendations. By reformulating the recommendation task as a text generation problem, these methods treat item identifiers as tokens within a vocabulary and fine-tune an LLM to predict the next item (Geng et al., 2022; Bao et al., 2023; Lu et al., 2024; Kim et al., 2024). This paradigm allows the model to harness the world knowledge embedded within the LLM. While these generative models have demonstrated impressive performance in capturing complex user preferences, their reliance on autoregressive generation results in significant inference latency, making them impractical for real-time applications that must serve millions of users.

To address the latency issue, another line of research utilizes LLMs as powerful feature encoders. In this approach, an LLM processes textual information associated with users or items (e.g., item descriptions, user reviews) to produce high-quality semantic embeddings (Liu et al., 2025; Sheng et al., 2025; Kim et al., 2025; Jia et al., 2025). Although this method is significantly faster at inference time, it treats the LLM as a static knowledge extractor. Consequently, it often fails to capture the dynamic, context-dependent reasoning that is a key advantage of LLMs, effectively using their knowledge but not their active reasoning process.

### 2.2 LLM DISTILLATION FOR RECOMMENDATION

To capture the best of both deep reasoning of LLMs and the efficiency of smaller models, knowledge distillation has emerged as a promising research direction (Gu et al., 2024; Panigrahi et al., 2025). These methods aim to transfer the capabilities of a teacher LLM to a smaller, faster student model. Distilling into SLMs can transfer nuanced reasoning, but the resulting autoregressive students remain too slow for real-time applications and risk propagating the teacher's misalignments (Wang et al., 2024a;b). Conversely, distilling into traditional, non-generative recommenders achieves low latency but fails to capture the internal reasoning process, mimicking only the final outputs (Wang et al., 2025; Cui et al., 2024). In contrast to these approaches, our work focuses on distilling the LLM's reasoning capabilities into a pretrained text encoder, aiming to achieve both the reasoning of an LLM and the efficiency of an embedding-based system.

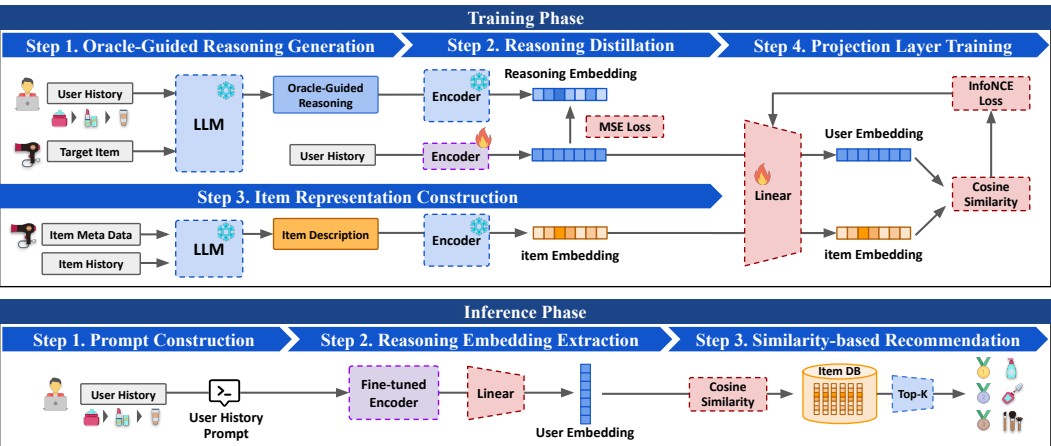

Figure 2: Overview of proposed method: **R2END**

# 3 REASONING-TO-ENCODER DISTILLATION

In this section, we introduce our proposed method, **R**easoning-**to-En**coder **D**istillation for Recommendation (**R2END**). Our method is designed to distill the high-fidelity, aligned reasoning of a LLM into a lightweight and efficient text encoder for scalable recommendation. The overall architecture follows a teacher-student paradigm, consisting of three main stages: (1) Offline generation of oracle-guided reasoning and rich item descriptions using a powerful teacher LLM; (2) Training a student text encoder to mimic the LLM's reasoning process through distillation; and (3) An LLM-free inference stage that relies solely on the fast student encoder for recommendation. Figure 2 provides an overview of our framework.

## 3.1 ORACLE-GUIDED REASONING GENERATION

A primary challenge in distilling LLM reasoning is the risk of learning from flawed or misaligned rationales generated by the teacher model. To mitigate this, we introduce an oracle-guided generation process to ground the LLM's reasoning in factual user behavior, ensuring explicit alignment with the ground truth. For a given user $u$, let their chronological interaction history be denoted as a sequence of items $S_u = (i_1, i_2, \ldots, i_n)$. The ground-truth next item that the user interacts with is $i_{n+1}$. This sequence is then verbalized into a natural language sentence to form the user's raw history text, $H_u$. We construct a prompt, $P_u$, that includes both the user's history and this ground-truth item $i_{n+1}$, which serves as the "oracle." The LLM is then tasked to generate a reasoning text, $R_u$, that explains why user $u$ would choose item $i_{n+1}$ given their past actions. This process can be formulated as:

$$R_u = \text{LLM}(P_u(H_u, i_{n+1})) \tag{1}$$

By providing $i_{n+1}$ as the oracle, we constrain the LLM's reasoning to be factually grounded, preventing it from generating speculative or incorrect rationales that diverge from the user's actual preferences. This ensures that the knowledge source for our distillation is of high quality and directly relevant to the recommendation task.

## 3.2 REASONING DISTILLATION TO TEXT ENCODER

The core of our framework is to distill the reasoning capability, now captured in the aligned text $R_u$, into a computationally efficient student text encoder, $\text{Enc}_S$. The student encoder's goal is to learn to produce a "reasoning-infused" embedding directly from the user's raw history text, $H_u$. To create reasoning-infused target embeddings, we employ a pretrained text encoder ($Enc$) to convert the textual rationales, previously generated by our teacher LLM, into dense vector representations. To further enrich the supervised signal, we concatenate the generated reasoning $R_u$ with the metadata of the ground-truth item, $M_{i_{n+1}}$ (e.g., title, category, brand). This combined text, $R'_u = R_u \oplus M_{i_{n+1}}$, is then encoded by the encoder to produce the target reasoning embedding, $e_u^R \in \mathbb{R}^d$:

$$e_u^R = Enc(R'_u). \tag{2}$$

The student encoder $\text{Enc}_S$ takes the user's history text as input and generates a corresponding user embedding, $e_u = \text{Enc}_S(H_u)$. We then train the student encoder by minimizing the Mean Squared Error (MSE) between its output and the teacher's target embedding. The distillation loss, $\mathcal{L}_{\text{distill}}$, is defined as:

$$\mathcal{L}_{\text{distill}} = \frac{1}{|\mathcal{U}|} \sum_{u \in \mathcal{U}} \|e_u - e_u^R\|_2^2, \tag{3}$$

where $\mathcal{U}$ is the set of all users in the training data. This objective forces the student encoder $\text{Enc}_S$ to internalize the semantic essence of the LLM's aligned reasoning process, enabling it to generate a user representation that simultaneously embeds both recommendation signals and the distilled reasoning.

### 3.3 ITEM REPRESENTATION CONSTRUCTION

To obtain high-quality item embeddings that reside in the same semantic space as our user embeddings, we leverage LLMs to generate rich, context-aware item descriptions. Standard item metadata is often insufficient to provide the necessary context or describe salient features for recommendation. We therefore aim to generate a richer item representation grounded in both the extensive world knowledge of an LLM and the contextual behavior of users who have previously purchased the item. To overcome this, we prompt the LLM to create a descriptive text $D_i$ for each item $i$, conditioning not only on its intrinsic metadata $M_i$ but also on the interaction histories of users who previously purchased it. This provides valuable context about the item's key features and appeal. The generated description $D_i$ is then encoded using the same pre-trained encoder $Enc$ to produce the final item embedding $e_i \in \mathbb{R}^d$:

$$D_i = \text{LLM}(P_i(M_i, H_i)) \tag{4}$$

$$e_i = \text{Enc}(D_i). \tag{5}$$

This ensures that both user and item embeddings are represented within a shared, meaningful semantic space, which is crucial for effective similarity-based recommendation.

### 3.4 TRAINING PROJECTION LAYER

To further strengthen the supervisory signal and explicitly optimize for the recommendation task, we introduce a contrastive learning objective. The reasoning-infused user embedding $e_u$ and the item embedding $e_i$ are passed through a shared projection layer, $f(\cdot)$, respectively, to map them into a shared latent space for dense retrieval. This process yields the final user representation $z_u$ and item representation $z_i$, which are optimized for the final similarity computation:

$$z_u = f(e_u), \ \ z_i = f(e_i). \tag{6}$$

We then employ the InfoNCE loss to maximize the similarity between a user and their ground-truth item (positive sample) while minimizing it for other items (negative samples). The contrastive loss is formulated as:

$$\mathcal{L}_{\text{InfoNCE}} = \sum_{u \in \mathcal{U}} -\log \frac{\exp(\cos(z_u, z_i^+)/\tau)}{\sum_{j \in \mathcal{I}_u^-} \exp(\cos(z_u, z_j)/\tau)}, \tag{7}$$

where $\mathcal{I}_u^-$ is the set of negative items of user $u$, $z_i^+$ is representation of a positive item. $\cos(\cdot)$ is cosine similarity, and $\tau$ is a temperature hyperparameter.

To stabilize training and prevent overfitting, we incorporate an $L_2$-based regularization term applied to all embeddings involved in the contrastive loss:

$$\mathcal{L}_{\text{reg}} = \lambda \left( \|z_u\|_2 + \|z_i^+\|_2 + \frac{1}{N} \sum_{j=1}^{N} \|z_j\|_2 \right), \tag{8}$$

where $\lambda$ is a regularization coefficient. The final training objective for projection layer is a weighted sum of the contrastive loss and regularization term:

$$\mathcal{L}_{\text{total}} = \mathcal{L}_{\text{InfoNCE}} + \mathcal{L}_{\text{reg}}. \tag{9}$$

## 3.5 LLM-FREE INFERENCE

One of the key advantages of our framework is its highly efficient and scalable LLM-free inference process, which is crucial for recommender systems. In contrast to prior works, we entirely exclude the LLM at inference time and instead rely solely on the student text encoder, which has been infused with the LLM's reasoning capabilities via our distillation process. For an incoming user request, their history text $H_u$ is first converted into an embedding $e_u$ by the encoder and then passed through a projection layer to produce the final user representation $z_u$. Recommendations are then conducted by ranking all candidate items in the corpus $\mathcal{I}$ based on the cosine similarity between the user representation $z_u$ and each pre-computed item representation $z_i$. The final set of top-K recommendations, $\mathcal{I}_K(u)$, is identified as follows:

$$\mathcal{I}_K(u) = \arg \operatorname*{top-k}_{i \in \mathcal{I}} \cos(z_u, z_i). \qquad (10)$$

This architectural choice not only bypasses the significant latency and computational costs of autoregressive LLMs but also decouples the complex reasoning generation from the real-time serving loop, resulting in a highly scalable and practical system for deploying reasoning-based recommendations.

## 4 EXPERIMENT

In this section, we present extensive experiments to demonstrate the effectiveness of R2END, aiming to answer the following research questions (**RQs**).

- **RQ1** Does R2END achieve state-of-the-art performance compared to existing LLM-based and distillation-based recommendation baselines?
- **RQ2** How significant are the improvements in inference latency and throughput offered by R2END when compared to existing distillation-based approaches?
- **RQ3** How do the core components of R2END contribute to its overall performance improvement?
- **RQ4** Does the student encoder's user embedding successfully capture the semantic essence of the teacher LLM's reasoning?
- **RQ5** Does the reasoning-infused embedding generated by R2END lead to more diverse and novel recommendations, particularly for less popular long-tail items, compared to existing models?

### 4.1 EXPERIMENTAL SETUP

In this subsection we describe experimental setups. In our experiments, we employ Gemma3 (Team et al., 2025) models of various scales: a 12B model serves as the teacher, a 1B model as the student, and a 4B model for the LLM-based recommendation baselines. A publicly available text encoder is utilized for both our proposed method and relevant baselines (Li & Li, 2024). More details and hyperparameters are described in Appendix and our online repository[1].

**Datasets.**   We conducted experiments on three widely used benchmark datasets : *Sports*, *Beauty*, and *Toys*. These datasets cover different domains, allowing us to evaluate the robustness and generalizability of our method in diverse domains.

**Evaluation Metrics.**   We adopt standard ranking metrics, including Hit Rate (HR) and Normalized Discounted Cumulative Gain (NDCG), to evaluate recommendation performance. Specifically, we report results at cut-off values of top-5 and top-10 (i.e., HR@5, NDCG@5, HR@10, NDCG@10). All evaluations are conducted over the full item pool, which consists of over 10K items per domain. This setup closely resembles real-world deployment scenarios and contrasts with prior LLM-based recommendation studies, which typically evaluate small-scale candidates.

**Baselines.**   To evaluate our model, we compare it with diverse baselines in three categories.

**(1) Conventional recommendation models.** We include representative recommendation baselines such as GRU4Rec (Hidasi, 2015), BERT4Rec (Sun et al., 2019), SASRec (Kang & McAuley, 2018), FDSA (Zhang et al., 2019), and S$^3$-Rec (Zhou et al., 2020), which have long been widely used for sequential recommendation by modeling users' sequential interaction patterns.

---

[1]https://anonymous.4open.science/r/R2END/

Table 1: Performance comparison of existing recommendation methods (Top-5 metrics). The best results for each metric are highlighted in bold, and the second-best results are underlined. "H@5" and "N@5" denote Hit Rate and NDCG at rank 5, respectively.

| Category | Method | LLM | Sports | | Beauty | | Toys | | Yelp | |
|---|---|---|---|---|---|---|---|---|---|---|
| | | | H@5 | N@5 | H@5 | N@5 | H@5 | N@5 | H@5 | N@5 |
| Conventional Method | GRU4Rec | - | 0.0129 | 0.0086 | 0.0164 | 0.0099 | 0.0097 | 0.0059 | 0.0152 | 0.0099 |
| | SASRec | - | 0.0233 | 0.0154 | 0.0387 | 0.0249 | 0.0463 | 0.0306 | 0.0223 | 0.0141 |
| | BERT4Rec | - | 0.0115 | 0.0075 | 0.0203 | 0.0124 | 0.0116 | 0.0071 | 0.0051 | 0.0033 |
| | FDSA | - | 0.0182 | 0.0122 | 0.0267 | 0.0163 | 0.0228 | 0.0140 | 0.0271 | 0.0170 |
| | S$^3$-Rec | - | 0.0251 | 0.0161 | 0.0387 | 0.0244 | 0.0443 | 0.0294 | 0.0168 | 0.0123 |
| LLM-based Method | AlphaRec (MLP) | Gemma3 (4B) | 0.0157 | 0.0099 | 0.0285 | 0.0183 | 0.0258 | 0.0174 | 0.0052 | 0.0024 |
| | AlphaRec (LGCN) | Gemma3 (4B) | 0.0210 | 0.0139 | 0.0280 | 0.0193 | 0.0107 | 0.0068 | 0.0044 | 0.0021 |
| | LLMEmb | Gemma3 (4B) | 0.0250 | 0.0160 | 0.0482 | 0.0310 | 0.0561 | 0.0369 | 0.0122 | 0.0076 |
| | LLM-SRec | Gemma3 (4B) | 0.0215 | 0.0101 | 0.0384 | 0.0237 | 0.0364 | 0.0225 | 0.0294 | 0.0173 |
| | LEARN | Gemma3 (4B) | 0.0115 | 0.0075 | 0.0157 | 0.0095 | 0.0213 | 0.0137 | 0.0047 | 0.0027 |
| Teacher | SLIM(T) | Gemma3 (12B) | 0.0273 | 0.0174 | 0.0452 | 0.0298 | 0.0524 | 0.0343 | 0.0491 | 0.0414 |
| Student | SLIM(S) | Gemma3 (1B) | 0.0247 | 0.0160 | 0.0419 | 0.0275 | 0.0499 | 0.0325 | 0.0486 | 0.0413 |
| | RDRec | T5-Large (0.7B) | 0.0045 | 0.0031 | 0.0162 | 0.0117 | 0.0053 | 0.0039 | 0.0114 | 0.0092 |
| | SLMRec (8→4) | Gemma3 (1.9B) | 0.0278 | 0.0162 | 0.0500 | 0.0308 | 0.0518 | 0.0321 | 0.0416 | 0.0303 |
| | DLLM2Rec | - | 0.0169 | 0.0104 | 0.0284 | 0.0174 | 0.0378 | 0.0248 | 0.0125 | 0.0080 |
| | RLMRec | - | 0.0302 | 0.0215 | 0.0357 | 0.0257 | 0.0141 | 0.0089 | 0.0133 | 0.0101 |
| Ours | R2SLM (SFT) | Gemma3 (1B) | 0.0226 | 0.0146 | 0.0464 | 0.0312 | 0.0529 | 0.0350 | 0.0518 | 0.0438 |
| | R2SLM (Logit KD) | Gemma3 (1B) | 0.0171 | 0.0113 | 0.0335 | 0.0218 | 0.0451 | 0.0289 | 0.0419 | 0.0332 |
| | R2SASRec | | 0.0185 | 0.0115 | 0.0291 | 0.0181 | 0.0411 | 0.0264 | 0.0106 | 0.0066 |
| | **R2END** | Text Encoder (0.3B) | **0.0344** | **0.0221** | **0.0664** | **0.0450** | **0.0712** | **0.0483** | **0.0595** | **0.0514** |
| Improvement | | | +13.91% | +2.79% | +37.76% | +45.16% | +26.92% | +30.89% | +21.18% | +24.15% |

**(2) LLM-based recommendation methods.** We include recent methods that utilize LLMs for retrieval over the entire item pool, such as AlphaRec (Sheng et al., 2025), LLMEmb (Liu et al., 2025), and LLM-SRec (Kim et al., 2025). We evaluate two variants of AlphaRec using LightGCN and MLP. For these baselines, we utilized a medium-sized LLM (4B). Since this approach does not involve distillation, our rationale was to select a model size that represents a fair middle ground between the larger teacher (12B) and smaller student (1B) models.

**(3) LLM distillation-based recommendation methods.** We compare our proposed method against state-of-the-art baselines that focus on distilling the reasoning capabilities of LLMs. Among these, SLIM (Wang et al., 2024b) and RDRec (Wang et al., 2024a) distill reasoning capabilities of a larger teacher LLM into a smaller student model. Additionally, we include SLMRec (Xu et al., 2025), which utilizes only a subset of an LLM's layers for distillation and integrates these layers with embeddings from a conventional recommender (e.g., SASRec).

## 4.2 OVERALL PERFORMANCE

We conducted experiments to evaluate our method against baselines to answer **RQ1**. To further validate the effectiveness of our proposed method, we also compared our primary approach of distilling reasoning into an encoder (R2END) against a variant that distills the same reasoning into a SLM (R2SLM). We used the generated reasoning text as the target for SFT training. Table 1 shows the overall recommendation performances. Our proposed method consistently outperforms all strong baselines, achieving an average performance improvement of 29% and up to 45% over the best baseline results. Furthermore, the encoder-based approach achieved superior performance over the SLM variant, which suggests that distilling into an encoder enables more stable and effective learning. Notably, our proposed method, R2END, surpasses the performance of the teacher model-based baseline (SLIM(T)). We also observe that our encoder-based variant is more effective than the SLM-based variant. These results indicate that our approach more effectively mitigates the misalignment between the LLM's general reasoning and the specific context of the recommendation domain. Furthermore, this validates the effectiveness of our proposed distillation strategy, which creates a "reasoning-infused" embedding space.

## 4.3 INFERENCE EFFICIENCY

To answer **RQ2**, we further analyze these practical advantages by comparing inference latency and throughput against various distillation-based baselines, as shown in Figure 3. The results clearly demonstrate that our approach is significantly more efficient than methods that distill reasoning into SLMs, highlighting its effectiveness for large-scale recommendation scenarios. Interestingly, while our method exhibits slightly higher latency and lower throughput than SLMRec, we recall from our

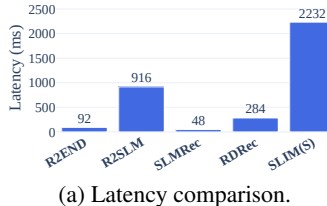 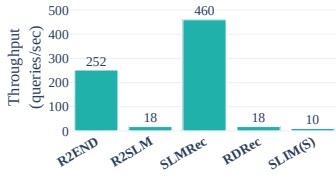

(a) Latency comparison.  (b) Throughput comparison.

Figure 3: Inference Efficiency Comparison. We compare the inference latency (ms, lower is better) and throughput (queries/sec, higher is better) against various distillation-based baselines. The results demonstrate that our proposed method achieves significantly higher inference efficiency.

Table 2: Ablation study results. The best results for each metric are highlighted in bold.

| Method | Sports | | | | Beauty | | | | Toys | | | | Yelp | | | |
|---|---|---|---|---|---|---|---|---|---|---|---|---|---|---|---|---|
| | H@5 | N@5 | H@10 | N@10 | H@5 | N@5 | H@10 | N@10 | H@5 | N@5 | H@10 | N@10 | H@5 | N@5 | H@10 | N@10 |
| **R2END** | **0.0344** | **0.0221** | **0.0517** | **0.0277** | **0.0664** | **0.0450** | **0.0958** | **0.0545** | **0.0712** | **0.0483** | **0.1035** | **0.0587** | **0.0595** | **0.0514** | **0.0701** | **0.0548** |
| (1) w.o. Oracle Guide (ALL) | 0.0307 | 0.0198 | 0.0464 | 0.0248 | 0.0549 | 0.0366 | 0.0806 | 0.0449 | 0.0603 | 0.0402 | 0.0893 | 0.0496 | 0.0476 | 0.0398 | 0.0557 | 0.0425 |
| (2) w.o. Oracle Guide (RS) | 0.0313 | 0.0199 | 0.0469 | 0.0249 | 0.0573 | 0.0381 | 0.0828 | 0.0463 | 0.0605 | 0.0407 | 0.0896 | 0.0502 | 0.0288 | 0.0206 | 0.0412 | 0.0246 |
| (3) w.o. LLM Reasoning | 0.0217 | 0.0135 | 0.0368 | 0.0183 | 0.0475 | 0.0309 | 0.0728 | 0.0390 | 0.0574 | 0.0377 | 0.0847 | 0.0464 | 0.0336 | 0.0251 | 0.0464 | 0.0292 |
| (4) w.o. Text Encoder | 0.0255 | 0.0163 | 0.0410 | 0.0213 | 0.0477 | 0.0307 | 0.0751 | 0.0395 | 0.0468 | 0.0308 | 0.0760 | 0.0402 | 0.0539 | 0.0452 | 0.0666 | 0.0493 |
| (5) w.o. Projection | 0.0176 | 0.0116 | 0.0269 | 0.0146 | 0.0367 | 0.0235 | 0.0538 | 0.0290 | 0.0511 | 0.0341 | 0.0752 | 0.0419 | 0.0364 | 0.0284 | 0.0456 | 0.0313 |
| (6) w.o. $\mathcal{L}_{reg}$ | 0.0320 | 0.0207 | 0.0506 | 0.0267 | 0.0616 | 0.0412 | 0.0914 | 0.0509 | 0.0671 | 0.0455 | 0.0989 | 0.0558 | 0.0579 | 0.0483 | 0.0699 | 0.0525 |

main performance evaluation that our approach achieved substantially higher accuracy. This suggests that utilizing an LLM as a simple embedding extractor, may limit its reasoning capabilities. While a trade-off between performance and efficiency is evident, our work strikes an effective balance, successfully leveraging the reasoning of LLMs while achieving a practical level of efficiency suitable for real-world deployment.

## 4.4 ABLATION STUDY

To answer **RQ3** and analyze the contribution of each key component in our method, we conduct a comprehensive ablation study. We systematically remove or replace core components of our model and observe the impact on performance. The results are summarized in Table 2.

**Effect of Oracle-Guided Reasoning.** We first investigate the critical role of our oracle-guided generation process. We compare our method with two variants: **(1) w.o. Oracle Guide (ALL)**, which uses all reasoning texts generated by the teacher LLM without the ground-truth guidance, and **(2) w.o. Oracle Guide (RS)**, rejection sampling approach, which only uses cases where the unguided LLM's recommendation successfully ranked the ground-truth item within the top-100. The results show that both variants underperform our method, with the rejection sampling version achieving slightly better results than using all unguided data. This finding strongly validates our hypothesis and underscores the importance of generating reasoning that is explicitly aligned with the ground-truth for effective distillation.

**Effect of LLM Reasoning and Text Encoder.** Second, we analyze the impact of the distilled reasoning and the encoder architecture itself. We test two variants: **(3) w.o. LLM Reasoning**, where the encoder training target is replaced with the ground-truth item embedding alone, and **(4) w.o. Text Encoder,** where the student encoder is replaced by a simple mean pooling of the user's historical item embeddings. Both configurations lead to a substantial drop in performance, with the removal of LLM reasoning causing the most significant degradation. This indicates that merely using the ground-truth item as a target is insufficient. It highlights how our proposed method effectively uses LLM reasoning as a semantic bridge between a user's history and the ground-truth item, which is crucial for enhancing recommendation performance.

**Effect of Projection Layer and Regularization.** Finally, we examine the contribution of the final projection. We test our model **(5) w.o. Projection**, which removes the projection layer, and **(6) w.o. $\mathcal{L}_{reg}$**, which removes the regularization term. Removing the projection layer led to a noticeable performance decrease, while the removal of the regularization term, also degraded performance, albeit to a lesser extent. This emphasizes the necessity of the projection layer and the regularization term for effectively learning a supervised signal tailored to the final recommendation task.

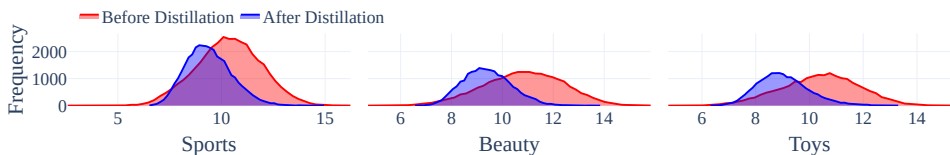

Figure 4: Comparison of L2 distance between user embeddings from encoder and oracle-guided reasoning embeddings, before and after distillation. A smaller distance indicates a higher similarity to the teacher's reasoning.

## 4.5 INDEPTH ANALYSIS

We conducted further analyses to provide deeper insights into the effectiveness and strengths of our proposed method. First, to answer **RQ4**, we investigate whether the student encoder successfully mimics the teacher LLM's reasoning by analyzing the changes in user representations before and after distillation. Second, to address **RQ5**, we evaluate our model's performance on recommendation diversity and its effectiveness on long-tail items, which are established strengths of LLM-based recommendation that we aim to preserve.

### 4.5.1 VERIFYING SEMANTIC ALIGNMENT OF REASONING-INFUSED EMBEDDINGS

To answer **RQ4**, we conducted an analysis to verify that our distillation process successfully imbues the student encoder with the semantic essence of the teacher LLM's reasoning. We measured the L2 distance between the student's generated user embeddings and the teacher's target reasoning embeddings, both before and after distillation on test set. The resulting distributions, visualized in Figure 4, clearly illustrate the effectiveness of our approach. The blue distribution, representing the reasoning-infused embeddings after distillation, shows a distinct shift towards higher similarity (lower L2 distance) with the teacher's embeddings compared to the pre-distillation state. This result confirms that our student encoder effectively learns to mimic the teacher's aligned reasoning process as intended.

### 4.5.2 LONG-TAIL PERFORMANCE

To address **RQ5**, we investigate our model's capacity for improving recommendation diversity by evaluating its performance on long-tail items. Previous studies have shown that leveraging LLMs enhances recommendation diversity and particularly alleviates the long-tail problem (Liu et al., 2024; 2025; Han et al., 2025). We define the long-tail set as the least popular 80% of items. As illustrated in Figure

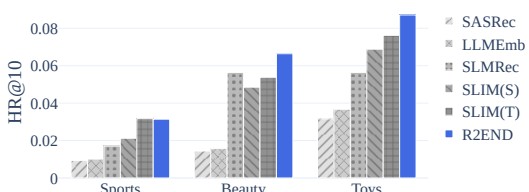

Figure 5: Hit@10 performance on long-tail items.

5, our proposed method, R2END, not only outperforms all baseline models but also surpasses the performance of the teacher model in this challenging setting. This result is particularly noteworthy as it demonstrates that our distillation framework successfully inherits the LLM's renowned strength in recommending diverse and less popular items, all while obviating the need for the LLM's computational overhead during inference.

## 5 CONCLUSION

In this work, we addressed the critical challenge of integrating the reasoning of LLMs into scalable recommender systems. We introduced Reasoning-to-Encoder Distillation, a novel framework that successfully distills high-fidelity, aligned reasoning into a lightweight text encoder. Our key innovation lies in the oracle-guided generation process, which grounds the LLM's rationale in actual user behavior, and our unique approach of "compiling" this complex reasoning into a single-vector representation. As demonstrated through extensive experiments, R2END not only achieves state-of-the-art performance but also drastically reduces inference latency. Ultimately, R2END offers a practical and effective pathway to building the next generation of recommender systems.

## ETHICAL STATEMENT

Our research aims to develop more efficient and accurate recommender systems, and we acknowledge the potential ethical implications inherent in such work. For our experiments, we utilized publicly available, anonymized benchmark datasets, mitigating direct privacy risks. However, we recognize that this data may contain inherent societal and demographic biases, which our method could learn and potentially amplify. Our analysis of long-tail performance is a deliberate step towards mitigating the "rich-get-richer" problem by improving recommendation diversity, but we acknowledge that this does not fully guarantee fairness across all user groups and item categories. Furthermore, like all recommender systems, the technology presented could be misused to create filter bubbles or for user manipulation. Our research focuses on the positive application of improving recommendation accuracy, and we are committed to transparency by releasing our code to allow the community to further investigate the behavior of our method. We believe that continued research into the fairness, transparency, and reliability of LLM-based recommender systems is essential.

## REPRODUCIBILITY STATEMENT

To ensure full reproducibility, we provide the complete source code in an anonymized online repository[2]. A detailed description of the datasets is available in Appendix A. All hyperparameters, and details of the computational environment are documented in Appendix C and the configuration files in our code repository. Finally, our evaluation protocol and the specific metrics used for performance comparison are described in the Experiments section.

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

APPENDIX

# A DATASETS

Table 3: Statistics of the datasets.

| Dataset | #Users | #Items | #Reviews | Density (%) |
|---------|--------|--------|----------|-------------|
| Sports  | 35,598 | 18,357 | 296,337  | 0.0453      |
| Beauty  | 22,363 | 12,101 | 198,502  | 0.0734      |
| Toys    | 19,412 | 11,924 | 167,597  | 0.0724      |
| Yelp    | 30,431 | 20,033 | 316,354  | 0.0519      |

We conducted experiments on three widely used benchmark datasets : *Sports*, *Beauty*, and *Toys*. These datasets cover different domains, allowing us to evaluate the robustness and generalizability of our method in diverse domains. Each dataset consists of user interactions, including a user ID, an item ID, a rating, a review, and a timestamp. The statistics of the dataset are provided in Table 3. These datasets are widely adopted by related studies (Geng et al., 2022; Rajput et al., 2023; Lee et al., 2025; Sun et al., 2024), which have been extensively explored over the past three years. We use five-core datasets, where both users and items have at least five interactions. For evaluation, we adopt the leave-one-out strategy, which is widely used in sequential recommendation research as a standard evaluation setup. This evaluation setup has been consistently adopted in previous studies (Kang & McAuley, 2018; Sun et al., 2019; Zhou et al., 2020; Li et al., 2023; Lee et al., 2025; Sun et al., 2024), ensuring comparability with existing methods.

# B BASELINES

To evaluate our method, we compare with a broad range of baseline models, grouped into three categories. Below, we briefly describe each baseline.

## B.1 CONVENTIONAL RECOMMENDATION METHOD

- **GRU4Rec** (Hidasi, 2015): One of the earliest sequential recommendation models, GRU4Rec employs gated recurrent units (GRUs) with a sequence-to-one pairwise ranking objective to capture temporal user behavior.

- **SASRec** (Kang & McAuley, 2018): Proposed to balance the efficiency of Markov Chains and the expressiveness of RNNs, SASRec uses self-attention to capture both short- and long-term user behavior. SASRec is designed to learn long-term user preferences based on only a small number of past actions by utilizing a self-attention mechanism. SASRec is a representative self-attention-based model.

- **BERT4Rec** (Sun et al., 2019): Designed to overcome the limitations of unidirectional models, BERT4Rec uses bidirectional self-attention to capture full sequence context. It employs a Cloze task to predict masked items, enabling richer sequence representations and improved performance across benchmarks.

- **FDSA** (Zhang et al., 2019): Aimed at capturing richer sequential patterns, FDSA models both item-level and feature-level transitions using separate self-attention blocks. By integrating heterogeneous item features and their dynamics, it improves recommendation performance over models that consider only item sequences.

- **$S^3$-Rec** (Zhou et al., 2020): To address data sparsity in sequential recommendation, $S^3$-Rec introduces self-supervised pre-training with four auxiliary objectives that capture correlations among items, attributes, and subsequences. By enhancing data representations through mutual information maximization, it achieves strong performance, especially under limited data scenarios.

## B.2 LLM-BASED RECOMMENDATION METHOD

- **AlphaRec** (Sheng et al., 2025): AlphaRec is a recommendation framework that challenges the necessity of traditional ID-based embeddings. Its core finding is that the rich represen-

tation space of a LLM already implicitly contains collaborative signals. AlphaRec builds a simple yet effective recommendation model directly on top of item embeddings extracted from a language model, demonstrating that this approach can outperform leading ID-based methods. For our experiments, we evaluate both the MLP and the Light Graph Convolutional Network (LGCN) variants of AlphaRec to ensure a comprehensive comparison.

- **LLMEmb** (Liu et al., 2025): LLMEmb leverages LLMs to generate semantically rich item embeddings, addressing the long-tail problem in sequential recommendation. Through supervised contrastive fine-tuning and recommendation adaptation training, LLMEmb aligns LLM-generated embeddings with collaborative signals, leading to performance gains across various sequential recommendation system models.

- **LLM-SRec** (Kim et al., 2025): To address the limited sequential understanding of LLM-based recommenders, LLM-SRec input user and item representations from a pre-trained sequential recommendation model into an LLM. This method achieved high performance by integrating the semantic information from the LLM with the CF signal from the CF-based model.

- **LEARN** (Jia et al., 2025): LEARN is a framework designed to synergize open-world knowledge from pre-trained LLMs with collaborative signals, aiming to overcome the semantic limitations of traditional ID-based embeddings. To address computational complexity, it employs a frozen LLM as an item encoder within a twin-tower architecture, effectively aligning textual semantics with user-item interactions while preventing catastrophic forgetting.

### B.3 LLM Distillation-based Recommendation Method

- **SLIM** (Wang et al., 2024b): This method proposes a LLM-based recommendation approach by distilling the reasoning capabilities of a large model into a smaller one. In our implementation, we utilize the Gemma3 12B model as the teacher and perform supervised fine-tuning on the Gemma3 1B student model using the generated reasoning text as the training data.

- **RDRec** (Wang et al., 2024a): This work addresses the problem that existing LLM-based recommenders do not explicitly learn the rationales behind user-item interactions. To solve this, a T5-based method where a smaller model learns by distilling rationales generated by a larger teacher LLM from user/item reviews. In our implementation, we utilize the Gemma3 12B model as the teacher and perform supervised fine-tuning on the T5-Large model as our student, using the generated reasoning text as the training data. We selected T5-Large as its parameter count is the most comparable to the other student models in our experiments. Furthermore, to mitigate the label leakage issue caused by token-level similarities in item IDs, as identified in recent studies (Lin et al., 2024), we adopted the sequential item ID assignment method from P5-SID for our evaluation (Hua et al., 2023).

- **SLMRec** (Xu et al., 2025): SLMRec proposes a method for faster and more efficient inference based on the empirical finding that many intermediate LLM layers are redundant for recommendation tasks. It uses knowledge distillation to transfer knowledge from a larger teacher model to a smaller student SLM. The architecture feeds embeddings from a traditional CF-based model into the LLM through an adapter. In our implementation, we designated 8 layers of a Gemma 12B model as the teacher and 4 layers as the student, incorporating item embeddings from SASRec. While this method is highly efficient, its reliance on CF-based embeddings and its failure to leverage the LLM's reasoning capabilities are significant limitations.

- **DLLM2Rec** (Cui et al., 2024): DLLM2Rec is a distillation strategy designed to transfer the capabilities of LLMs into lightweight conventional sequential models, thereby addressing inference latency constraints. To tackle challenges such as unreliable teacher knowledge and the capacity gap between models, it employs an importance-aware ranking distillation mechanism that filters and weights knowledge based on teacher confidence and student-teacher consistency. Additionally, it incorporates collaborative embedding distillation to integrate semantic knowledge from LLM embeddings with collaborative signals.

- **RLMRec** (Ren et al., 2024): RLMRec is a framework designed to enhance existing collaborative filtering models by integrating LLM-empowered representation learning. Address-

ing the limitations of ID-based recommenders and the noise inherent in implicit feedback, it incorporates auxiliary textual signals and employs an LLM-based profiling paradigm to capture complex user preferences.

## C  IMPLEMENTATION DETAILS

The text generation process was implemented using vLLM[3], while the embedding extraction was based on SentenceTransformers[4] library. We used the Gemma 3 (Team et al., 2025) model for LLM-based generation and *mxbai-embed-large-v1* (Li & Li, 2024) as the text encoder in the main experiments. All experiments were carried out on a single NVIDIA RTX A6000 GPU with 40GB of VRAM in Ubuntu 22.04.3 LTS environment.

Our mehtod is trained in two main stages. For the initial reasoning distillation, we fine-tune the student text encoder for a single epoch with a batch size of 16 and a learning rate of $1e - 5$. For the subsequent recommendation task training, the projection layer is trained for 10 epochs using a contrastive objective with 99 negative samples per positive instance. We set the temperature for contrastive loss to 0.07. In this stage, we use a batch size of 128, a learning rate of $1e - 4$, and an output embedding dimension of 512. The regularization weight $\lambda$ is set to 0.5. All experiments are conducted with a fixed random seed of 22 for reproducibility. A comprehensive list of all hyperparameters and other implementation details is provided in our publicly available repository.

## D  ADDITIONAL EXPERIMENTS AND ANALYSIS

### D.1  PERFORMANCE OF TEACHER LLM REASONING

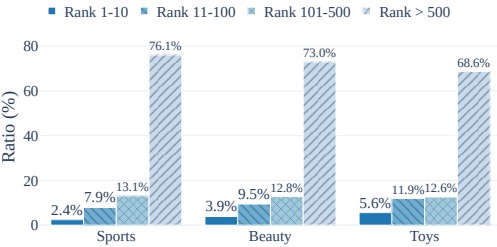

(a) Recommendation performance of the teacher LLM (Gemma3-12B) with step-by-step reasoning.

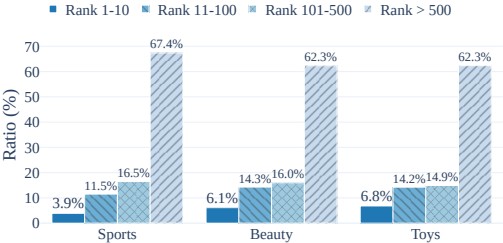

(b) Recommendation performance of the teacher LLM (Gemma3-27B) with step-by-step reasoning.

Figure 6: Misalignment of unguided LLM reasoning. The LLM-generated rationale, without knowing the ground-truth item, frequently diverges from actual user behavior. This underscores the necessity of oracle guidance to ensure the reasoning is aligned and suitable for distillation.

Our research originates from the observation that even with their advanced reasoning capabilities, LLMs struggle to identify the correct ground-truth item from a large candidate pool in recommendation tasks. To substantiate this, Figure 6 presents the performance of applying the Gemma3 12B and 27B models to the SLIM baseline, a state-of-the-art reasoning-based method. Despite the substantial size of these models, both achieve a top-10 hit rate of less than 7% across all datasets. This result provides strong empirical evidence for the gap we claim exists between the general-purpose reasoning of LLMs and the specific, contextual nature of actual user behavior in the recommendation domain.

### D.1.1  VERIFYING SEMANTIC ALIGNMENT OF REASONING-INFUSED EMBEDDINGS

While the improvements in recommendation accuracy are a clear benefit, we also sought to verify that the distillation process was successful in its primary goal: ensuring the student encoder's embeddings capture the semantic essence of the LLM's reasoning. To this end, we analyzed the

---

[3]https://docs.vllm.ai/
[4]https://sbert.net/

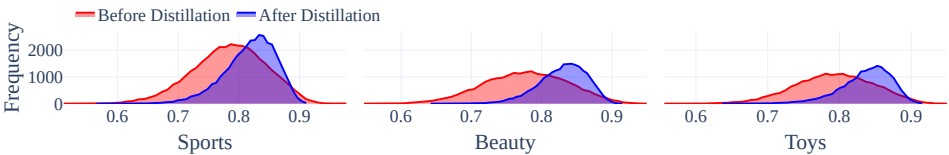

Figure 7: Comparison of cosine similarity between user embeddings from encoder and oracle-guided reasoning embeddings, before and after distillation. A larger value indicates a higher similarity to the teacher's reasoning.

cosine similarity between the user embeddings generated by the student encoder and the target reasoning embeddings from the teacher, comparing the distributions before and after distillation. The results, illustrated in Figure 7, show a significant increase in cosine similarity across all datasets. This provides strong evidence that our framework not only enhances recommendation accuracy but also successfully mimics the teacher's aligned reasoning as intended.

### D.2 PERFORMANCE COMPARISON BY STUDENT ENCODER

Table 4: Performance Comparison of Different Student Encoders.

| Encoder | Sports | | | | Beauty | | | | Toys | | | | Yelp | | | |
|---|---|---|---|---|---|---|---|---|---|---|---|---|---|---|---|---|
| | H@5 | N@5 | H@10 | N@10 | H@5 | N@5 | H@10 | N@10 | H@5 | N@5 | H@10 | N@10 | H@5 | N@5 | H@10 | N@10 |
| Best Baseline | 0.0278 | 0.0174 | 0.0433 | 0.0225 | 0.0482 | 0.0310 | 0.0767 | 0.0398 | 0.0561 | 0.0369 | 0.0838 | 0.0458 | 0.0491 | 0.0414 | 0.0588 | 0.0437 |
| mxbai-embed-large (335M) | 0.0344 | 0.0221 | 0.0517 | 0.0277 | 0.0664 | 0.0450 | 0.0958 | 0.0545 | 0.0712 | 0.0483 | 0.1035 | 0.0587 | 0.0595 | 0.0514 | 0.0701 | 0.0548 |
| mxbai-embed-xsmall (24M) | 0.0268 | 0.0176 | 0.0406 | 0.0220 | 0.0500 | 0.0324 | 0.0735 | 0.0399 | 0.0592 | 0.0396 | 0.0864 | 0.0484 | 0.0532 | 0.0474 | 0.0597 | 0.0495 |
| bge-large-en (335M) | 0.0303 | 0.0192 | 0.0473 | 0.0247 | 0.0512 | 0.0335 | 0.0780 | 0.0421 | 0.0658 | 0.0450 | 0.0959 | 0.0546 | 0.0575 | 0.0495 | 0.0674 | 0.0527 |
| gte-base-en-v1.5 (100M) | 0.0317 | 0.0215 | 0.0468 | 0.0263 | 0.0586 | 0.0402 | 0.0847 | 0.0486 | 0.0717 | 0.0502 | 0.0959 | 0.0580 | 0.0513 | 0.0429 | 0.0622 | 0.0464 |
| Qwen3-embedding (0.6B) | 0.0288 | 0.0184 | 0.0449 | 0.0236 | 0.0530 | 0.0348 | 0.0787 | 0.0431 | 0.0623 | 0.0418 | 0.0902 | 0.0508 | 0.0513 | 0.0431 | 0.0624 | 0.0466 |

To investigate the generalization capability of our proposed method, we conducted experiments using various text encoders as the student model. The results, presented in Table 4, show that our approach generally outperforms the baselines regardless of the specific encoder used. This demonstrates that our Reasoning-to-Encoder Distillation framework is broadly applicable to diverse encoder architectures. However, we observed that a very small model with only 24M parameters underperformed the best of baselines. This suggests that a certain model capacity is necessary to effectively internalize the complex reasoning distilled from the LLM. Nevertheless, our method surpassed other strong LLM-based approaches with an encoder of just 335M parameters, highlighting its high efficiency and broad applicability.

## E PROMPT FORMATS

In this section, we provide a more detailed explanation of the prompt design of oracle-guided reasoning and item descriptions used in our experiments.

### E.1 ORACLE-GUIDED REASONING PROMPT

To generate the oracle-guided reasoning, we construct a detailed prompt for the teacher LLM. Figure 8 illustrates the prompt format used for the reasoning generation. This prompt provides the model with the user's historical purchase records, which are constructed using up to their 8 most recent interactions from the last 60 days. If a user has no purchases within this period, we use their single most recent transaction. Each interaction in the history text includes the item's purchase date and its associated metadata. Crucially, the prompt also includes the ground-truth next item, which serves as the "oracle." This setup compels the LLM to generate reasoning that is explicitly aligned with the user's actual behavior within the recommendation domain. We instruct the LLM to generate this reasoning within a maximum length of 512 words and reduce randomness by setting the temperature to zero. When embedding this reasoning with the text encoder, we incorporate metadata from the ground-truth item to better reflect recency.

---
**Oracle-Guided Reasoning Generation Prompt**

### Task:
Based on the user's chronological purchase history and the target item, analyze and summarize user's preferences related to the target item.
Focus on identifying the user's preferences related to the target item from the user's purchase history.

### User's Purchase History:
{user_purchase_history}

### Target Item:
{target_item}

### Requirements:
- Start with "The user's preferences related to the target item are as follows:"
- Write a single coherent paragraph (max {max_words} words) summarizing the user's preferences related to the target item.

### Response:

---

Figure 8: Oracle-guided reasoning generation prompt.

---
**Item Description Generation Prompt**

### Task:
Analyze the provided metadata and reviews to determine what kinds of users are most likely to prefer the target item. Your response should start with: "Users who prefer [common themes/preferences] would find this item suitable." Incorporate patterns from the reviews, such as favored features, usage scenarios, or functional benefits. Avoid generic statements and ensure your description is grounded in the review content.

### Target Item Metadata:
- **Target Item Title**: {item_title}
- **Brand**: {brand}
- **Original Description**: {description}

### Reviews of Previously Purchased Items:
{previous_item_reviews}

### Requirements:
- Begin with the sentence: "Users who prefer ..."
- Use a single paragraph, no more than {max_words} words.
- Focus on inferred user traits, preferences, and realistic use cases.

### Response:

---

Figure 9: Item description generation prompt.

### E.2 ITEM DESCRIPTION PROMPT

Figure 9 presents the prompt format used for item description generation. Instead of relying on static metadata, our description is generated by leveraging the one-step-prior interactions (at time $T-1$) from the users who purchased that item. This process aims to construct item description based on users behavior, thereby capturing the user preference that leads to a purchase. We set the maximum output length to 512 words and reduce randomness by fixing the temperature at zero. When embedding the final item representation using the text encoder, we concatenate the generated description with the item's metadata to preserve both semantic and factual aspects. For this item description generation, we used Gemma3-4B, same model used in LLM-based baselines.

## F LLM USAGE DISCLOSURE

We used LLMs solely as auxiliary tools during the writing process of this paper. Specifically, LLMs were employed to help with improving the clarity of sentences, polishing grammar, and suggesting alternative phrasings for better readability. The research ideas, methodology, experimental design, implementation, analysis, and conclusions were entirely conceived and executed by the authors. No parts of the technical content, including theoretical results, models, algorithms, or experiments, were generated by LLMs. The role of LLMs was limited to text editing assistance, similar in scope to grammar checkers or writing support tools.

