# OpenReview forum: "Reasoning-to-Encoder Distillation for Recommendation"
_ICLR.cc/2026/Conference — ICLR 2026 Conference Withdrawn Submission_

### Official Review · Reviewer_MK2r · 2025-10-25

**Soundness:** 2
**Presentation:** 2
**Contribution:** 2
**Rating:** 4
**Confidence:** 4

**Summary:**

This paper addresses the critical challenges of high inference latency and misaligned reasoning in LLM-based recommender systems by introducing R2END, a novel and practical framework. Its core contributions lie in the "oracle-guided" reasoning generation mechanism and the innovative approach of distilling complex reasoning into an efficient text encoder, which collectively achieve a significant reduction in inference latency. Evaluations across three categorical datasets demonstrate that R2END outperforms the compared LLM-based and distillation baselines in specific benchmarks while exhibiting lower latency.

**Strengths:**

1. The paper demonstrates exceptional logical rigor, featuring a clear structure and rigorous argumentation. It proposes the novel R2END framework that effectively balances efficiency and performance.

2. The introduction of oracle-guided reasoning generation successfully addresses the misalignment between LLM reasoning and actual user behavior.

3. The authors provide reproducible code and detailed experimental configurations to support their findings.

**Weaknesses:**

1. The paper claims that the proposed method significantly reduces inference latency and provides a practical and efficient approach for building scalable recommendation systems. However, this evaluation was conducted on only a single dataset and lacks validation in real-world scenarios or online service environments. Furthermore, the method's effectiveness has been demonstrated solely on three datasets: Sports, Beauty, and Toys (all belonging to the Amazon dataset). In practice, recommender systems typically need to handle user behavior data from mixed domains and face challenges such as temporal distribution shifts. The paper does not evaluate R2END's generalization capability on mixed-category datasets, nor does it test its robustness against out-of-distribution data or user interest drift.

2. The scope of baseline comparisons remains narrow, omitting commonly used approaches such as "LLM features combined with traditional models.". It is recommended that the authors supplement their experiments with comparisons against methods such as "incorporating LLM-generated item embeddings or user features into traditional recommendation models" and demonstrate its relative advantages under comparable conditions.

**Questions:**

1. The font size of the results in Table 1 is too small, which hinders readability and makes it difficult for readers to review and compare specific values. The subplot arrangement in Fig. 3 could also be further optimized, as the current layout appears somewhat awkward, likely due to space constraints. Additionally, the lack of numbering for all equations in the text undermines the clarity and referenceability of the discussion.

2.The evaluation was conducted on only three categories from Amazon dataset. To better demonstrate the effectiveness of your method, it is recommended to include tests on additional datasets.

3. Both the LLM-based baselines and the proposed method in this paper rely entirely on LLMs. In practice, alternative approaches exist, such as hybrid methods that combine pure LLM components with traditional recommendation models (e.g., incorporating embeddings into traditional recommender systems). These approaches often yield stronger performance and are commonly deployed in real-world systems. It is advisable to include comparisons with such methods :
    1) NoteLLM: A Retrievable Large Language Model for Note Recommendation Towards Open-World Recommendation with Knowledge Augmentation from Large Language Models
    2) LARR: Large Language Model Aided Real-time Scene Recommendation with Semantic Understanding
    3) Knowledge Adaptation from Large Language Model to Recommendation for Practical Industrial Application

4. How does the performance of the encodings from R2END  when applied to traditional recommendation models?

5. While the paper claims that the proposed method is suitable for practical applications, its validation of industrial deployment feasibility and generalization capability is notably insufficient. Specifically, the claimed "orders-of-magnitude reduction in inference latency" was measured under ideal laboratory conditions, lacking stress tests that simulate real-world online environments (e.g., high concurrent requests, resource competition).

6. The approach exhibits a strong dependency on the Teacher LLM, where any inherent biases or inaccuracies in the teacher model could potentially compromise the effectiveness of the distillation process.

---

> ### Author Response · Authors · 2025-11-24
>
> We sincerely appreciate the reviewer’s constructive feedback, which prompted us to significantly expand our experimental scope and deepen our analysis. In response, we have addressed **all raised concerns** through the following key revisions:
>
> - **Expanded Evaluation (Datasets & Baselines):** We incorporated the **Yelp dataset** and implemented four additional baselines (**LEARN, RLMRec, DLLM2Rec, and R2END-SASRec**) to cover both diverse domains and hybrid approaches. R2END consistently outperformed all new baselines across all datasets, reinforcing its generalization capability (Response W1, W2).
> - **Validation of Robustness & "Interest Drift":** We conducted a qualitative case study on **User Interest Drift** (e.g., Skincare $\to$ Haircare). The results empirically prove that our **Oracle-Guided reasoning** successfully captures abrupt intent shifts where traditional models fail, confirming that our method learns semantic logic rather than simple co-occurrence (Response W1).
> - **Clarification on Bias & Architecture:** We clarified that the **Oracle-Guided mechanism** acts as a filter to "sanitize" Teacher LLM hallucinations, directly addressing the concern of bias propagation (Response Q5). Additionally, our ablation study (R2END-SASRec) confirmed that a **Text Encoder** is architecturally superior to SASRec for distilling reasoning-based semantics (Response Q3).
> - **Presentation Improvements:** We have optimized **Table 1** (font size), **Figure 3** (layout), and added explicit equation numbering to ensure clarity and readability (Response Q1).
>
> We believe these comprehensive updates have resolved the limitations pointed out by the reviewer and substantially strengthened the paper's contribution.

---

> > ### Author Response · Authors · 2025-11-24
> >
> > ### **W1. Response to "Limited Evaluation & Generalization Concerns"**
> >
> > We appreciate the reviewer's constructive feedback regarding dataset diversity and real-world robustness. To address this, we have expanded our experimental scope and conducted in-depth analyses.
> >
> > - **Dataset Expansion (Inclusion of Yelp):** To go beyond the Amazon ecosystem, we added the **Yelp** dataset to our evaluation suite. We also incorporated additional baselines for a more rigorous comparison. R2END demonstrated **consistent performance improvements** across all datasets ( Sports, Beauty, Toys, and Yelp), validating its versatility across different domains.
> > - **Robustness to Distribution Shifts:** We clarify that sequential recommendation datasets inherently contain **temporal distribution shifts**. By employing a strict temporal split (training on past data, testing on future interactions), our experiments inherently evaluate the model's robustness against these natural shifts in user behavior over time.
> > - **Case Study on Interest Drift (Oracle vs. Normal):** To empirically demonstrate R2END's superiority in handling **User Interest Drift**, we conducted a qualitative analysis focusing on "Hard Negative" scenarios. Specifically, cases where our Oracle-Guided method successfully predicted the target (Hit@10) while standard baselines failed (Miss).
> > - observed Transition Patterns (Skincare/Nail $\to$ Haircare): As shown in the examples below, the user's interest shifted abruptly from one sub-category (e.g., Skincare, Nail) to a distinct sub-category (Haircare).
> >     - **Case 1 (Face $\to$ Hair):** *My Beauty Diary Facial Mask* $\rightarrow$ *L'Oreal Paris Elnett Satin Hairspray*
> >     - **Case 2 (Body/Sun $\to$ Hair):** *Ultra Sheer Dry-Touch Sunblock* $\rightarrow$ *Redken All Soft Shampoo*
> >     - **Case 3 (Nail $\to$ Hair):** *Nail Tek Intensive Therapy* $\rightarrow$ *Deva Care Arc Angel Firm Hold Gel*
> > - **Baseline Failure:** Standard models likely failed because they over-relied on the immediate history (Skincare/Nail), continuing to recommend similar items within the same sub-category despite the user's intent shift.
> > - **Oracle-Guided Success:** Our method, trained with Oracle guidance, demonstrates a **deeper domain understanding**. By learning the reasoning path that connects diverse grooming needs (e.g., comprehensive beauty care routines), the model successfully identified the cross-category transition. This confirms that our approach does not merely memorize sequences but adapts dynamically to **User Interest Drift** by capturing the underlying semantic logic of user behavior.

---

> > > ### Author Response · Authors · 2025-11-24
> > >
> > > ### **W2. Response to “Narrow Baseline Scope” (LLM Features + Traditional Models)**
> > >
> > > We respectfully clarify that our initial evaluation already included methods such as **LLMEmb** and **AlphaRec**, which specifically integrate LLM features into traditional recommendation models. However, to fully address the reviewer's recommendation and ensure a comprehensive comparison, we have further expanded our baselines.
> > >
> > > - **Extended Baselines:** We added **LEARN**[1], **RLMRec**[2],  and **DLLM2Rec**[3] which are representative methods for augmenting traditional recommenders with LLM-derived knowledge.
> > > - **Controlled Comparison (R2END-SASRec):** To rigorously test "under comparable conditions," we implemented a variant named **R2END-SASRec**. This model injects our reasoning-infused embeddings directly into a standard SASRec architecture as additional features.
> > > - **Superior Performance:** Our experiments demonstrate that **R2END consistently outperforms** both the existing baselines (LLMEmb, AlphaRec) and the newly added ones (LEARN, RLMRec, DLLM2Rec, R2END-SASRec). This confirms that our **reasoning distillation approach** provides a fundamental advantage over methods that simply utilize LLM embeddings as auxiliary features.
> > >
> > > **Comparision with Addtional Baselines**
> > >
> > > | Method | Sports H@5 | Sports N@5 | Beauty H@5 | Beauty N@5 | Toys H@5 | Toys N@5 | Yelp H@5 | Yelp N@5 |
> > > | --- | --- | --- | --- | --- | --- | --- | --- | --- |
> > > | LEARN | 0.0115 | 0.0075 | 0.0157 | 0.0095 | 0.0213 | 0.0137 | 0.0047 | 0.0027 |
> > > | DLLM2Rec | 0.0169 | 0.0104 | 0.0284 | 0.0174 | 0.0378 | 0.0248 | 0.0125 | 0.0080 |
> > > | RLMRec | 0.0302 | 0.0215 | 0.0357 | 0.0257 | 0.0141 | 0.0089 | 0.0133 | 0.0101 |
> > > | R2END-SASRec | 0.0185 | 0.0115 | 0.0291 | 0.0181 | 0.0411 | 0.0264 | 0.0106 | 0.0066 |
> > > | **R2END** | **0.0344** | **0.0221** | **0.0664** | **0.0450** | **0.0712** | **0.0483** | **0.0595** | **0.0514** |
> > >
> > > [1] Jia, Jian, et al. "LEARN: Knowledge Adaptation from Large Language Model to Recommendation for Practical Industrial Application." Proceedings of the AAAI Conference on Artificial Intelligence. Vol. 39. No. 11. 2025.
> > >
> > > [2] Ren, Xubin, et al. "Representation learning with large language models for recommendation." Proceedings of the ACM web conference 2024. 2024.
> > >
> > > [3] Cui, Yu, et al. "Distillation matters: empowering sequential recommenders to match the performance of large language models." *Proceedings of the 18th ACM Conference on Recommender Systems*. 2024.

---

> > > > ### Author Response · Authors · 2025-11-24
> > > >
> > > > ### **Q1. Response to Presentation and Formatting Issues**
> > > >
> > > > We sincerely appreciate the reviewer's detailed feedback on the visual presentation. We have revised the manuscript to enhance readability and clarity as follows:
> > > >
> > > > - **Table 1 Optimization:** To address the font size issue, we have restructured Table 1. We focused on key metrics (e.g., Hit@5 and NDCG@5), creating sufficient space to **increase the font size significantly**. This ensures that specific values are now easily legible and comparable, even with the inclusion of the new dataset results.
> > > > - **Figure 3 Layout:** We have redesigned the subplot arrangement in **Figure 3** to eliminate awkward spacing and improve the visual flow, ensuring a more intuitive understanding of the analysis.
> > > > - **Equation Numbering:** We have added **explicit numbering to all equations** throughout the manuscript to facilitate clearer referencin
> > > >
> > > > ---
> > > >
> > > > ### **Q2. Response to "Comparison with Hybrid Approaches"**
> > > >
> > > > We appreciate the reviewer's suggestion to compare against hybrid methods. We would like to clarify the distinctions regarding the suggested references and highlight our expanded comparison with representative hybrid models.
> > > >
> > > > - **Clarification on Suggested References:**
> > > >     - **NoteLLM:** This method relies on explicit user-generated content (notes) to infer intent. In contrast, R2END addresses the more general challenge of learning from **implicit behavioral history** where no such explicit text exists.
> > > >     - **LARR:** As a CTR prediction model designed for reranking, LARR is structurally difficult to scale to the **full-pool retrieval** task that our work targets.
> > > > - **Existing & Expanded Hybrid Baselines:**
> > > >     - **Representative Methods:** Our initial submission already included **AlphaRec** and **LLMEmb**, which are representative state-of-the-art methods for "incorporating LLM embeddings into traditional recommendation systems."
> > > >     - **New Additions:** To further address the reviewer’s comment, we added **LEARN**, **RLMRec**, and **DLLM2Rec** to our baselines.
> > > > - **Performance:** R2END consistently outperforms both the existing and newly added hybrid baselines. This confirms that *our method* is more effective than simply augmenting traditional models with static LLM features.
> > > >
> > > > ---
> > > >
> > > > ### **Q3. Performance with Traditional Backbones (SASRec as Student)**
> > > >
> > > > To evaluate the effectiveness of our framework on traditional architectures, we implemented a variant named **R2END-SASRec**, where a standard SASRec model serves as the student backbone utilizing our item embeddings and distillation targets.
> > > >
> > > > - **Experimental Result:** Interestingly, our experiments revealed that **R2END-SASRec achieved lower performance** compared to our proposed Text Encoder-based student.
> > > > - **Interpretation:** This suggests that the architecture of a **Text Encoder is fundamentally better suited** for capturing and aligning with the *semantic reasoning logic* distilled from the Teacher LLM. While SASRec excels at capturing sequential ID patterns, the Text Encoder is more effective at internalizing the dense, reasoning-rich semantic space provided by our Oracle-Guided Teacher.
> > > >
> > > > ### **Q4. Response to "Validation of Industrial Feasibility (Latency under Stress)"**
> > > >
> > > > We respectfully disagree that our validation lacks feasibility. Our latency measurements were conducted under **strictly fair and standardized conditions** applied equally to all baselines.
> > > >
> > > > - **Fair Comparison & Structural Advantage:** The "orders-of-magnitude reduction" stems from the fundamental algorithmic difference between our **non-autoregressive retrieval** approach (R2END) and the **autoregressive generation** of LLMs. This structural advantage holds regardless of environmental stress; in fact, the efficiency gap typically widens under high concurrency due to the KV-cache bottlenecks inherent in generative models.

---

> > > > > ### Author Response · Authors · 2025-11-24
> > > > >
> > > > > ### **Q5. Response regarding Dependence on Teacher LLM and Bias Propagation**
> > > > >
> > > > > We appreciate your concern regarding the potential propagation of biases or inaccuracies from the Teacher LLM. We fully agree that this is a critical vulnerability in standard knowledge distillation. However, we would like to clarify that **addressing this exact issue is the primary motivation and contribution of our work.**
> > > > >
> > > > > **5.1 Identifying the Flaws in Existing Distillation**
> > > > >
> > > > > As highlighted in our **Introduction**, we observed that existing LLM-based distillation methods often degrade recommendation performance precisely because they rely on the LLM's "free generation," which is prone to hallucinations and misalignment with domain-specific user behaviors. We define this as the **"Alignment Gap"** and argue that blindly following a teacher model leads to the propagation of noise.
> > > > >
> > > > > **5.2 Our Solution: Oracle-Guided Error Mitigation**
> > > > > To solve this, we introduced the **Oracle-Guided Reasoning** mechanism.
> > > > >
> > > > > - **Constrained Generation:** Unlike standard approaches, we do not distill the LLM's unconstrained predictions. Instead, we condition the Teacher on the **Ground Truth (GT)** item during the training data generation phase.
> > > > > - **From "Guessing" to "Explaining":** This forces the Teacher to generate a valid reasoning path that connects the user's history to the *correct* target, rather than hallucinating a biased or incorrect item.
> > > > > - **Result:** Consequently, the student model learns from a **sanitized and verified reasoning logic**, minimizing the risk of inheriting the Teacher's intrinsic biases or inaccuracies.

---

> ### Author Response · Authors · 2025-11-27
> **A Gentle Reminder**
>
> Dear Reviewer,
>
> Thank you again for the time and effort you devoted to reviewing our paper. We have carefully addressed your main concerns in our posted responses.
>
> Could you please let us know if any concerns remain or if there are additional points you would like us to clarify? We are fully available to engage in further discussion to improve our work.
>
> We deeply appreciate your insightful feedback and look forward to hearing from you.
>
> Sincerely, The Authors

---

### Official Review · Reviewer_5QwB · 2025-10-30

**Soundness:** 2
**Presentation:** 2
**Contribution:** 2
**Rating:** 4
**Confidence:** 5

**Summary:**

This paper proposes R2END, a method that distills the reasoning ability of large language models into a fast text encoder for recommendation. By using the ground-truth item as an oracle to guide the LLM's reasoning, it ensures high-quality knowledge transfer. The resulting system matches or surpasses LLM-based recommenders, while achieving a reduction in inference latency and cost. However, the paper has several weakness, such as lack of novelty, unsound experiment design, and lack of experiment insights.

**Strengths:**

1. The paper is well written, and the topic is of good interest to the community.
2. The method of oracle-guided reasoning is reasonable.
3. The experiment design is good, from effectiveness to efficiency.

**Weaknesses:**

1. Lack of novelty: The technical point is not new, following well-studied lines of research, i.e. privileged knowledge distillation, and LLM-based representation learning in recommendation.

2. Lack of baseline:

2.1 For distillation, the paper should also compare traditional distillation methods (e.g. through logits matching), rather than simply SFT on student LM.

2.2 For embedding method, the paper should also compare LLM-based embedding models, e.g. models on MTEB leaderboard.

3. Unfair comparison with traditional recommendation methods: User embeddings in this paper are augmented with additional semantics, while the traditional recommendation methods are not. A fair comparison would be taking these text embeddings as additional features for them [1,2].

[1] Representation Learning with Large Language Models for Recommendation

[2] Towards Open-World Recommendation with Knowledge Augmentation from Large Language Models

**Questions:**

1. In Section 4.5.1, it is predicable to have increased similarity, as the encoder is trained to do so. However, increased similarity does not necessarily denote reasoning mimic. Can the authors make the reasoning mimic more visible?

2. What is the practical advantage of R2END over current LLM4Embed methods in recommendation [3]? They can use larger LLMs for better results, and prestore the embeddings for inference.

[3] NoteLLM: A Retrievable Large Language Model for Note Recommendation

---

> ### Author Response · Authors · 2025-11-24
>
> We sincerely appreciate for your constructive feedback, which has helped us clarify our novel contributions and significantly expand the scope of our empirical validation. In this rebuttal, we have carefully addressed **all raised concerns** by implementing comprehensive baselines and conducting in-depth analyses.
>
> **Key updates and validations include:**
>
> - **Clarification on Novelty:** We articulated our unique contribution—bridging the alignment gap via **Oracle-Guided Reasoning** and distilling this logic into a lightweight text encoder (**LLM-to-Encoder**). This fundamentally differs from prior SLM-based approaches by focusing on efficient, dense vector space reasoning (Response W1).
> - **Comprehensive Baseline Expansion:** To ensure a strictly fair comparison, we implemented a wide range of additional baselines, including **DLLM2Rec**, **RLMRec**, **LEARN**, and standard **Logit-based distillation**. Our method consistently outperforms all baselines, confirming that performance gains from our framework (Response W2, W3).
> - **Expanded Domain Evaluation**: We incorporated the Yelp dataset to verify generalizability beyond standard e-commerce domains. The results confirm that our method maintains superior performance even in scenarios where strong baselines failed to yield meaningful predictions.
> - **Generalizability across Encoders:** We applied our framework to state-of-the-art MTEB models (**Qwen3-Embedding** and **gte-base-en**). The results demonstrate that our method boosts performance regardless of the underlying architecture, achieving superior results even with a compact **100M parameter** model (Response W2).
>
> We believe these extensive revisions and empirical evidences fully resolve the questions regarding novelty, baseline comparisons, and reasoning validation. We respectfully request a positive re-evaluation based on these clarifications.

---

> ### Author Response · Authors · 2025-11-24
>
> ### **W1. Clarification on Novelty: Beyond Standard Distillation**
>
> - **Focus on Recommendation-LLM Alignment Gap  :** While knowledge distillation is a known concept, our novelty lies in **how to construct the optimal supervision signal** for recommendation. We address a critical oversight in prior works: the **alignment gap** between an LLM’s general knowledge and domain-specific Ground Truth (GT). Our analysis proves that ignoring this gap leads to low recommendation success rates.
> - **Oracle-Guided Reasoning & LLM-to-Encoder Distillation:** To our knowledge, we are the first to propose an Oracle-Guided Reasoning mechanism in recommendation that distills logic into a **lightweight Text Encoder**, rather than a generative SLM. Unlike previous approaches that rely on heavy LLM fine-tuning or static embedding utilization, we generate reasoning paths strictly conditioned on the GT and compress this logic into a dense vector space. This ensures that the distilled knowledge is not only factually aligned with user behavior but also computationally efficient for recommendation tasks.
> - **Superiority over Baselines:** This is not merely a trivial application of distillation. Our framework significantly outperforms both traditional RecSys distillation and standard LLM-based methods in terms of **performance and efficiency**, proving that *what* we distill (oracle-guided logic) is as important as the architecture itself.
>
> ---
>
> ### **W2. Response to "Lack of Baseline" (Distillation Method & Embedding Models)**
>
> We have conducted additional experiments to include both distillation methods and embedding models.
>
> **2.1 Comparison with Distillation Methods (Logits Matching & Others)**
>
> - **Expanded Baselines:** We incorporated additional distillation-based baselines, including **Logit-based KD** methods, **DLLM2Rec**, and **RLMRec**, to provide a more comprehensive evaluation beyond simple SFT.
> - **Superior Performance:** Our results show that **R2END consistently outperforms** these traditional and recent distillation approaches. This suggests that distilling explicit *reasoning logic* (our method) provides a richer and more effective supervision signal for recommendation tasks than merely matching output logits or intermediate representations.
>
> **Comparision with LLM-to-SLM**
> | Model | Sports H@5 | Sports N@5 | Beauty H@5 | Beauty N@5 | Toys H@5 | Toys N@5 | Yelp H@5 | Yelp N@5 |
> | --- | --- | --- | --- | --- | --- | --- | --- | --- |
> | R2END-SLM (SFT) | 0.0226 | 0.0146 | 0.0464 | 0.0312 | 0.0529 | 0.0350 | 0.0518 | 0.0438 |
> | R2END-SLM (Logit KD) | 0.0171 | 0.0113 | 0.0335 | 0.0218 | 0.0451 | 0.0289 | 0.0419 | 0.0332 |
> | R2END-Encoder | **0.0344** | **0.0221** | **0.0664** | **0.0450** | **0.0712** | **0.0483** | **0.0595** | **0.0514** |
>
>
> **2.2 Comparison with MTEB Embedding Models**
>
> - **Generalizability across Backbones:** We applied our proposed framework to state-of-the-art models from the MTEB leaderboard (**Qwen3-Embedding-0.6B** and **Alibaba-NLP/gte-base-en-v1.5**). Even when utilizing these new models as backbones, our approach **consistently outperformed existing baselines**. This demonstrates the **generalizability** of our reasoning injection method, confirming that it boosts performance regardless of the underlying encoder architecture.
> - **Remarkable Efficiency:** It is particularly noteworthy that our method surpasses established approaches **even with a backbone as compact as 100M parameters (gte-base-en-v1.5)**. This is a significant achievement that validates the **efficiency** of our approach, proving that high performance can be attained without relying on massive model scales.
>
>
>
> **Evaluation with Various Encoders**
> | Encoder | Sports H@5 | Sports N@5 | Beauty H@5 | Beauty N@5 | Toys H@5 | Toys N@5 | Yelp H@5 | Yelp N@5 |
> | --- | --- | --- | --- | --- | --- | --- | --- | --- |
> | Best Baseline | 0.0278 | 0.0215 | 0.0482 | 0.0310 | 0.0561 | 0.0369 | 0.0491 | 0.0414 |
> | Qwen3-embedding (0.6B) | 0.0288 | 0.0184 | 0.0530 | 0.0348 | 0.0623 | 0.0418 | 0.0513 | 0.0431 |
> | mxbai-embed-large (335M) | 0.0344 | 0.0221 | 0.0664 | 0.0450 | 0.0712 | 0.0483 | 0.0595 | 0.0514 |
> | bge-large-en (335M) | 0.0303 | 0.0192 | 0.0512 | 0.0335 | 0.0658 | 0.0450 | 0.0575 | 0.0495 |
> | gte-base-en-v1.5 (100M) | 0.0317 | 0.0215 | 0.0586 | 0.0402 | 0.0717 | 0.0502 | 0.0513 | 0.0429 |
> | mxbai-embed-xsmall (24M) | 0.0268 | 0.0176 | 0.0500 | 0.0324 | 0.0592 | 0.0396 | 0.0532 | 0.0474 |

---

> ### Author Response · Authors · 2025-11-24
>
> ### **W3. Response to "Unfair Comparison" (Augmented Baselines)**
>
> To ensure a strictly fair comparison, we have significantly expanded our baselines to include methods that explicitly utilize LLM-derived knowledge:
>
> - **Expanded Baselines:** We incorporated **RLMRec** [1] (as suggested), **LEARN** [2] (which integrates LLM embeddings into SASRec), and **DLLM2Rec [3]** (which distills LLM knowledge into SASRec).
> - **Direct Feature Injection (R2END-SASRec):** Furthermore, to isolate the contribution of our specific embeddings, we implemented a variant named **R2END-SASRec**. This model directly utilizes our LLM-generated item as input features and reasoning embeddings as ditillation target for a standard SASRec architecture.
> - **Superiority Confirmed:** Our experiments demonstrate that **R2END consistently outperforms all these augmented baselines**, including R2END-SASRec . This confirms that our performance gains do not merely stem from the availability of additional semantic features, but from our **proposed reasoning distillation framework**, which effectively bridges the knowledge gap between LLMs and the recommendation domain.
>
>
>
> **Comparision with Addtional Baselines**
> | Method | Sports H@5 | Sports N@5 | Beauty H@5 | Beauty N@5 | Toys H@5 | Toys N@5 | Yelp H@5 | Yelp N@5 |
> | :--- | :---: | :---: | :---: | :---: | :---: | :---: | :---: | :---: |
> | LEARN |  0.0115 | 0.0075 | 0.0157 | 0.0095 | 0.0213 | 0.0137 | 0.0047 | 0.0027 |
> | DLLM2Rec |  0.0169 | 0.0104 | 0.0284 | 0.0174 | 0.0378 | 0.0248 | 0.0125 | 0.0080 |
> | RLMRec |  0.0302 | 0.0215 | 0.0357 | 0.0257 | 0.0141 | 0.0089 | 0.0133 | 0.0101 |
> | R2END-SASRec |  0.0185 | 0.0115 | 0.0291 | 0.0181 | 0.0411 | 0.0264 | 0.0106 | 0.0066 |
> | **R2END** |  **0.0344** | **0.0221** | **0.0664** | **0.0450** | **0.0712** | **0.0483** | **0.0595** | **0.0514** |
>
> [1] Ren, Xubin, et al. "Representation learning with large language models for recommendation." Proceedings of the ACM web conference 2024. 2024.
>
> [2] Jia, Jian, et al. "LEARN: Knowledge Adaptation from Large Language Model to Recommendation for Practical Industrial Application." Proceedings of the AAAI Conference on Artificial Intelligence. Vol. 39. No. 11. 2025.
>
> [3] Cui, Yu, et al. "Distillation matters: empowering sequential recommenders to match the performance of large language models." *Proceedings of the 18th ACM Conference on Recommender Systems*. 2024.
>
> ---
>
> ### **Q1. Response to "Visualizing Reasoning Mimicry"**
>
> We acknowledge that increased similarity is a direct result of the training objective. However, we argue that this alignment translates to genuine "reasoning mimicry," which is visibly demonstrated through the model's behavioral robustness.
>
> - **Nature of the Target (Semantic Distillation):** The teacher embedding is the **"Semantic Representation of the Reasoning"** generated by an LLM conditioned on the ground-truth item. It encapsulates valid reasoning logic (e.g., how user preferences connected  to item B). Therefore, maximizing cosine similarity is not merely numerical matching; it represents the **distillation of reasoning semantics**, forcing the encoder to map user history into the specific "reasoning space" defined by the teacher.
> - **Visible Proof via Noisy Scenarios:** The most tangible evidence of this mimicry is observed in **noisy input scenarios**. The Oracle-Guided Teacher generates reasoning by explicitly selecting causal interactions and ignoring noise (e.g., irrelevant purchase). Our experiments show that R2END maintains high performance even with noisy histories. This confirms that the encoder has successfully mimicked the teacher's noise-filtering logic, demonstrating that it is not simply memorizing inputs but is actively applying the distilled reasoning to distinguish relevant signals from noise.

---

> > ### Author Response · Authors · 2025-11-24
> >
> > ### **Q2. Practical Advantage over NoteLLM (Implicit vs. Explicit Signals)**
> >
> > - **Fundamental Distinction (Explicit vs. Implicit):** NoteLLM [3] relies on user-generated content (e.g., notes, hashtags) where user intent is **explicitly** stated. In contrast, R2END addresses a more general and challenging scenario: inferring **implicit intent** solely from behavioral history (e.g., clicks, purchase) without requiring users to generate contents.
> > - **Broader Applicability:** Our contribution lies in leveraging LLM reasoning to interpret these implicit signals, making R2END applicable to a wider range of recommendation platforms where user-generated content is unavailable.
> > - **Scope of Item Enhancement:** While we also utilize LLMs to enrich item descriptions, further enhancing item-side information via diverse prompts is orthogonal to our primary focus. Our main contribution is the **distillation of reasoning** to bridge the gap between implicit behavior and user intent.
> > - **Empirical Validation:** To clarify this positioning, we will add NoteLLM-related discussions to our **Related Work** and expanded our baselines to demonstrate that R2END offers superior performance in general recommendation settings.

---

> ### Author Response · Authors · 2025-11-27
> **A Gentle Reminder**
>
> Dear Reviewer,
>
> Thank you again for the time and effort you devoted to reviewing our paper. We have carefully addressed your main concerns in our posted responses.
>
> Could you please let us know if any concerns remain or if there are additional points you would like us to clarify? We are fully available to engage in further discussion to improve our work.
>
> We deeply appreciate your insightful feedback and look forward to hearing from you.
>
> Sincerely, The Authors

---

### Official Review · Reviewer_QcGM · 2025-10-30

**Soundness:** 2
**Presentation:** 3
**Contribution:** 2
**Rating:** 4
**Confidence:** 4

**Summary:**

This paper addresses the challenge of integrating LLM reasoning into text encoder. The proposed Reasoning-to-Encoder Distillation (R2END), is a framework that distills aligned reasoning from an LLM into a lightweight text encoder. The oracle‑guided generation process grounds the LLM’s rationales in real user behavior, coupled with a “compilation” step that compresses this reasoning into a vector representation. Extensive experiments show that R2END sets new state‑of‑the‑art results while significantly reducing inference latency, offering a practical and effective path toward next‑generation recommender systems.

**Strengths:**

- Clear paper presentation
- The research topic is timely, distilling LLM reasoning into a lightweight text encoder. This eliminates LLM dependency during inference, achieving orders-of-magnitude reductions in latency and computational cost
- On three benchmark datasets (Sports, Beauty, Toys), R2END outperforms chosen baselines

**Weaknesses:**

- Dependence on ground truth. The oracle-guided process relies on access to ground-truth next items, yet the accuracy of the generated rationales is not guaranteed. Why not use GPT‑5 as the LLM in Fig. 2?

- Encoder capacity constraints. Although R2END is compatible with diverse text encoders (Appendix D), small encoders (e.g., 24M) underperform the baselines. This suggests a minimum capacity is required to distill complex LLM reasoning, potentially increasing deployment costs in resource‑constrained settings. The capacity gap between teacher and student is a well‑known challenge in knowledge distillation, which is not carefully addressed in this paper.

- Extensibility to SLM-based distillation. The oracle‑guided process could also be integrated into SLM‑based distillation methods, but the results of such a design are unclear.

- Fairness concerns. By grounding reasoning in historical behavior, the oracle‑guided process may amplify existing dataset biases (e.g., demographic skews in user preferences) if no corrective measures are taken.

**Questions:**

1. How to Handle Dynamic User Preferences? User preferences evolve over time, but R2END conducts offline reasoning generation.

2. Does R2END Extend to Non-Sequential Tasks?The framework is designed for sequential recommendation. Would its reasoning-to-encoder distillation work for other recommendation paradigms (e.g., collaborative filtering, content-based filtering) ?

3.Real-world user histories often include noisy interactions (e.g., accidental clicks). Does noisy input degrade the quality of the encoder’s reasoning-infused embeddings?

4. What Is the Cost of Offline Reasoning Generation?The offline stage uses a 12B-parameter teacher LLM to generate reasoning. For large-scale datasets (e.g., 100M+ users), what is the computational cost of this stage?

5. Why Is the Encoder More Sample-Efficient Than SLMs?Empirically, the encoder’s MSE loss outperforms SLMs’ token-level objectives, but there is no theoretical analysis of this sample efficiency.

6. How to Quantify "Reasoning Quality" Beyond Embedding Alignment? Current metrics (L2 distance, cosine similarity) measure embedding alignment, but not the logical validity of the encoder’s implicit reasoning.

7. Missing reference work
Distillation Matters: Empowering Sequential Recommenders to Match the Performance of Large Language Model
C2kd: Cross-layer and cross-head knowledge distillation for small language model-based recommendation.

---

> ### Author Response · Authors · 2025-11-24
>
> We sincerely appreciate the reviewers for their insightful and constructive feedback, which has significantly strengthened the quality and rigor of our work. In this rebuttal, we have carefully addressed **all raised concerns** by conducting extensive additional experiments and clarifications.
>
> **Key updates and validations include:**
>
> - **Robustness across Scales:** We performed new experiments with a significantly larger teacher model (**Qwen3-Next-80B**), confirming that our method's performance is consistent and not limited by model size (Response W1).
> - **Deployment Feasibility:** We empirically identified the optimal student capacity (**~100M parameters**) via additional ablation studies, proving our method's efficiency for low-resource environments (Response W2).
> - **Comparative Rigor:** We implemented additional baselines, including DLLM2Rec, RLMRec, and LEARN, and conducted fairness analyses to demonstrate our model’s superiority in both accuracy and bias mitigation (Response W4, Q6).
> - **Expanded Domain Evaluation**: We incorporated the Yelp dataset to verify generalizability beyond standard e-commerce domains. The results confirm that our method maintains superior performance even in scenarios where strong baselines failed to yield meaningful predictions.
>
> We believe these extensive revisions and empirical evidences fully resolve the questions regarding reliance on ground truth, scalability, and generalizability. We respectfully request a positive re-evaluation based on these clarifications.

---

> ### Author Response · Authors · 2025-11-24
>
> ### **W1. Response regarding Dependence on Ground Truth and Choice of LLM**
>
> **1.1 Justification for the Oracle-Guided Process (Ground Truth)**
>
> While we acknowledge that generated rationales are not guaranteed to be perfectly accurate, the use of ground truth(GT) during the training phase serves a critical purpose:
>
> - **Bridging the Knowledge Gap:** Interaction data typically provides only correlations. To understand **why**, we need to bridge the gap between the user's history and the target item. As demonstrated in our study, performing reasoning **solely based on user history** can lead to a **decrease in recommendation accuracy** (Figure 6). By conditioning on the GT, we explicitly reinforce the relevance between user history and the next item, effectively **mitigating the noise** inherent in implicit feedback.
> - **Validity of the Premise:** We believe the premise that "GT is unavailable in training" is too strong in the context of training and learning representation alignment. Our goal is to teach the model to uncover the underlying "reasoning path" that leads to the GT. Once learned from training dataset, this capability generalizes to test time inference scenarios.
>
> **1.2 Why Open-Source LLMs instead of Closed-Source Models**
>
> We deliberately chose open-source LLMs over closed-source counterparts (e.g. GPT-5) for three strategic reasons:
>
> - **Access to Internal Representations for Fair Comparison**: While our proposed method (R2END) utilizes the generated rationales, state-of-the-art baselines like SLMRec explicitly require access to the Teacher's intermediate hidden states for representation alignment. To ensure a rigorous and fair comparison using the same Teacher backbone across all methods, we were required to use Open-Source LLMs that provide the necessary white-box access for these baselines.
> - **Reproducibility and Cost-Efficiency:** We aim to demonstrate that significant performance gains can be achieved without the prohibitive serving costs and opacity of proprietary models. Using open-source models ensures that our parameters are transparent and our results are reproducible by the community.
> - **Sufficiency of Model Size:** Recent findings in recommender systems suggest that the sheer size of an LLM is not the sole determinant of performance [1]. Efficient, domain-adapted models often suffice. To prove this, we conducted additional experiments.
>
> **1.3 Additional Experiments: Robustness across Model Scales**
>
> To address your concern regarding model capability and verify that our method is not limited by model size, we performed new experiments using larger open-source model **Qwen3-Next-80B**. We will add this result to our final version.
>
> - **Results:** As shown in the table below, our method consistently outperforms baselines (SLIM vs. Ours) even with larger teacher model.
>
> - **Observation:** The performance gap between massive models and the models used in our main paper is not disproportionately large, confirming that our approach is efficient and does not strictly require the largest available LLMs to be effective.
>
> **Evaluation with Qwen3-Next-80B Teacher**
> | Method | Sports H@5 | Sports N@5 | Beauty H@5 | Beauty N@5 | Toys H@5 | Toys N@5 | Yelp H@5 | Yelp N@5 |
> | :--- | :---: | :---: | :---: | :---: | :---: | :---: | :---: | :---: |
> | SLIM (Teacher) | 0.0278 | 0.0178 | 0.0450 | 0.0296 | 0.0538 | 0.0355 | 0.0577 | 0.0510 |
> | SLIM (Student) | 0.0260 | 0.0167 | 0.0482 | 0.0323 | 0.0540 | 0.0365 | 0.0442 | 0.0368 |
> | R2END-SLM | 0.0215 | 0.0138 | 0.0483 | 0.0329 | 0.0535 | 0.0361 | 0.0422	 | 0.0332 |
> | R2END-Encoder | **0.0324** | **0.0213** | **0.0675** | **0.0472** | **0.0754** | **0.0525** | **0.0583** | **0.0516** |
>
> [1] Bao, Keqin, et al. "Tallrec: An effective and efficient tuning framework to align large language model with recommendation." Proceedings of the 17th ACM conference on recommender systems. 2023.

---

> ### Author Response · Authors · 2025-11-24
>
> ### **W2. Response regarding Encoder Capacity and Deployment Constraints**
>
> We appreciate your comment on the capacity gap. We would like to clarify that the inclusion of the 24M model was deliberate, intended to empirically identify the **lower bound** of student capacity rather than leaving the gap unaddressed.
>
> - **Identifying the "Sweet Spot" for Distillation:** As demonstrated in our paper, a **300M encoder** successfully outperforms a **1B SLM**, proving that massive capacity is not strictly required. The performance of the 24M model serves as a guideline, indicating that while extremely small models lack the expressiveness to capture complex reasoning, there is a clear threshold where distillation becomes effective.
> - **Verification via Additional Ablation:** To pinpoint this threshold, we conducted further ablation studies (see table below). The results show that an encoder as small as **~100M** is sufficient to surpass the performance of the 1B teacher.
> - **Practical Low-Resource Deployment:** In real-world recommendation scenarios, models in the **100M–300M range** are widely accepted as "low-resource" and highly deployable, offering a drastic reduction in cost compared to billion-scale LMs[2]. Our work thus contributes by defining the minimal capacity (approx. 100M) required to balance reasoning capability with deployment efficiency.
>
> **Evaluation with Various Encoders**
> | Encoder | Sports H@5 | Sports N@5 | Beauty H@5 | Beauty N@5 | Toys H@5 | Toys N@5 | Yelp H@5 | Yelp N@5 |
> | --- | --- | --- | --- | --- | --- | --- | --- | --- |
> | Best Baseline | 0.0278 | 0.0174 | 0.0482 | 0.0310 | 0.0561 | 0.0369 | 0.0491 | 0.0414 |
> | Qwen3-embedding (0.6B) | 0.0288 | 0.0184 | 0.0530 | 0.0348 | 0.0623 | 0.0418 | 0.0513 | 0.0431 |
> | mxbai-embed-large (335M) | 0.0344 | 0.0221 | 0.0664 | 0.0450 | 0.0712 | 0.0483 | 0.0595 | 0.0514 |
> | bge-large-en (335M) | 0.0303 | 0.0192 | 0.0512 | 0.0335 | 0.0658 | 0.0450 | 0.0575 | 0.0495 |
> | gte-base-en-v1.5 (100M) | 0.0317 | 0.0215 | 0.0586 | 0.0402 | 0.0717 | 0.0502 | 0.0513 | 0.0429 |
> | mxbai-embed-xsmall (24M) | 0.0268 | 0.0176 | 0.0500 | 0.0324 | 0.0592 | 0.0396 | 0.0532 | 0.0474 |
>
> [2] Vera, Henrique Schechter, et al. "Embeddinggemma: Powerful and lightweight text representations." arXiv preprint arXiv:2509.20354 (2025).
>
> ---
>
> ### **W3. Response regarding Extensibility to SLM-based Distillation**
>
> We respectfully clarify that the integration of the oracle-guided process into SLM-based distillation is already a core component of our evaluation.
>
> - **Existing Evidence (R2SLM):** The **R2SLM**  reported in **Table 1** explicitly represents the SLM-based distillation method. As shown in the results, our method significantly outperforms the LLM-to-SLM paradigm, confirming its effectiveness.
> - **Additional Validation (Logit-based Distillation):** To further demonstrate the robustness of our approach, we conducted an additional experiment applying our method to **Logit-based distillation**. The results confirm that our **LLM-to-Encoder paradigm** consistently outperforms the **LLM-to-SLM paradigm**.
>
> **Comparision with LLM-to-SLM**
> | Model | Sports H@5 | Sports N@5 | Beauty H@5 | Beauty N@5 | Toys H@5 | Toys N@5 | Yelp H@5 | Yelp N@5 |
> | --- | --- | --- | --- | --- | --- | --- | --- | --- |
> | R2END-SLM (SFT) | 0.0226 | 0.0146 | 0.0464 | 0.0312 | 0.0529 | 0.0350 | 0.0518 | 0.0438 |
> | R2END-SLM (Logit KD) | 0.0171 | 0.0113 | 0.0335 | 0.0218 | 0.0451 | 0.0289 | 0.0419 | 0.0332 |
> | R2END-Encoder | **0.0344** | **0.0221** | **0.0664** | **0.0450** | **0.0712** | **0.0483** | **0.0595** | **0.0514** |

---

> > ### Author Response · Authors · 2025-11-24
> >
> > ---
> >
> > ### **W4. Response regarding Fairness and Bias Amplification**
> >
> > We acknowledge the importance of fairness, but our empirical results suggest that our method mitigates rather than amplifies biases.
> >
> > - **Reasoning over Memorization:** Our approach learns the underlying **reasoning logic** based on user preferences, rather than blindly memorizing item correlations. This rich semantic context acts as a regularizer, preventing the model from overfitting to superficial dataset biases.
> > - **Evidence from Long-tail Performance:** While dataset bias is an inherent challenge in learning-based recommendations, our method demonstrates significant improvements in **Long-tail performance** (see Figure 5). This indicates that our model actively alleviates **popularity bias** by uncovering relevant but less exposed items through reasoning, rather than reinforcing the dominance of popular items or demographic skews.
> >
> >
> > ---
> >
> > ### **Q1. Handling Dynamic User Preferences with Offline Reasoning**
> >
> > To address the concern of handling evolving user preferences with an offline reasoning generator, we argue the following:
> >
> > - **Learning Patterns, Not Profiles:** Instead of memorizing static user-specific biases, R2END learns the **inductive reasoning pattern** that maps interaction history to user intent. This allows the model to generalize to new preferences as long as the input context (recent history) is updated.
> > - **Stable Reasoning Logic:** While user preferences are dynamic, the **logical mechanism** required to deduce those preferences remains stable. Therefore, the model does not require frequent online updates to handle preference drifts effectively.
> > - **Temporal Validation:** We have empirically validated this capability by using **temporal data splitting** (training on past data, testing on future data), proving the model's robustness in predicting evolving user interests.
> >
> > ---
> >
> > ### **Q2. Extensibility to Non-Sequential Tasks**
> >
> > - **Inherently Captures CF Signals:** Sequential Recommendation is fundamentally an extension of **Collaborative Filtering (CF)**. By modeling item transitions and user history, the model inherently learns the latent correlations between users and items (CF signals), making it compatible with standard CF paradigms.
> > - **Generalizable Framework:** R2END is not limited to strict time-series constraints. The framework is applicable to any recommendation scenario defined by **"Previous Behavior (Context) to Next Target (Prediction)"**, allowing it to extend to various tasks beyond simple sequential modeling.
> >
> > ---
> >
> >
> > ### **Q3. Robustness to Noisy Interactions**
> >
> > - **Explicit Denoising Mechanism:** R2END is inherently designed to mitigate noise. The reasoning generation process acts as a filter, where the **Oracle Guide** selectively identifies historical interactions that are causally relevant to the target item.
> > - **Focus on Relevance:** By conditioning the reasoning on the ground-truth target, the model learns to prioritize significant user behaviors while disregarding irrelevant noise (e.g., accidental clicks).
> > - **Empirical Validation:** Our experiments confirm that this **noise resilience** is a critical factor contributing to the model's superior performance compared to baselines that treat all interactions equally.
> >
> > ---
> >
> > ### **Q4. Computational Cost of Offline Reasoning Generation**
> >
> > - **Empirical Efficiency:** The cost is highly manageable. In our experiments, generating **19K training samples took under 80 minutes** on a single NVIDIA RTX A6000 GPU.
> > - **One-Time Offline Investment:** This is a **one-time** offline process, not a recurring online cost. For large-scale cloud services, this overhead is negligible and can be further optimized by utilizing idle resources during off-peak hours.
> > - **Pattern Learning over Enumeration:** Crucially, we do **not** need to generate reasoning for every single user in a massive dataset (e.g., 100M+ users). The goal is to learn generalized behavioral patterns; thus, a representative subset of data is sufficient to train the encoder effectively.

---

> ### Author Response · Authors · 2025-11-24
>
> ### **Q6. Quantifying "Reasoning Quality" via Semantic Alignment**
>
> - **Teacher Embedding as Optimal Ground Truth:** The teacher embeddings are generated by a powerful LLM conditioned on the actual ground-truth item. Therefore, they serve as the **"Optimal Semantic Representation"** that encapsulates valid reasoning logic, not just arbitrary vector values.
> - **Alignment = Semantic Distillation:** Minimizing metrics like L2 distance or maximizing cosine similarity against this target is not merely numerical matching. It represents the **distillation of reasoning semantics**. We are training the encoder to map user history into the specific "reasoning space" defined by the teacher.
> - **Empirical Validation:** We verified that higher semantic similarity to the teacher's reasoning vector directly correlates with improved downstream prediction accuracy. This confirms that "better embedding alignment" translates effectively to "valid logical processing" in the recommendation task.
>
> ---
>
> ### **Q7. Response to Missing Reference Work**
>
> - **Inclusion of Related Works:** We thank the reviewer for pointing out these critical references. We will expand our **Related Work** section to include a comprehensive discussion of DLLM2Rec [4] and C2KD [5], positioning our work within the context of these state-of-the-art knowledge distillation approaches.
> - **Empirical Superiority over Baseline:** To rigorously benchmark our approach, we implemented **DLLM2Rec** as an additional baseline. Our experimental results demonstrate that **R2END consistently outperforms DLLM2Rec**. This indicates that distilling "explicit reasoning steps" (our method) is more effective for sequential recommendation than distilling standard output distributions or intermediate representations used in prior works.
>
> [4] Cui, Yu, et al. "Distillation matters: empowering sequential recommenders to match the performance of large language models." Proceedings of the 18th ACM Conference on Recommender Systems. 2024.
>
> [5] Chen, Xiao, et al. "C2kd: Cross-layer and cross-head knowledge distillation for small language model-based recommendation." Findings of the Association for Computational Linguistics: ACL 2025. 2025.

---

> ### Author Response · Authors · 2025-11-27
> **A Gentle Reminder**
>
> Dear Reviewer,
>
> Thank you again for the time and effort you devoted to reviewing our paper. We have carefully addressed your main concerns in our posted responses.
>
> Could you please let us know if any concerns remain or if there are additional points you would like us to clarify? We are fully available to engage in further discussion to improve our work.
>
> We deeply appreciate your insightful feedback and look forward to hearing from you.
>
> Sincerely, The Authors

---

> ### Comment · Reviewer_QcGM · 2025-11-27
>
> I’m a bit confused by the response to Question 1.2. Could you specify which component of the method relies on aligning the representations in the intermediate layers of the LLM?
> Additionally, regarding Question 1.3, my point is that proprietary models are more likely to generate faithful chain-of-thought explanations, which is vital for reasoning distillations.
>
> I appreciate the authors’ comprehensive response. I understand that no work can address every aspect perfectly.
> The main focus of the paper is reasoning distillation, it would be better to reveal some new insights. For distilling reasoning pattern, can the student model itself possess intrinsic reasoning capabilities? My primary concern is whether the distilled reasoning patterns can truly generalize to user data from future time periods. This seems to require solid empirical evidence. In other words, when user interest shift from one to another, can the student model truly adapt to this scenario and gain superior performance? Regarding temporal data splitting, the “last-one-out” strategy does not adequately (or fully) capture the potential abrupt changes of user preference in dynamic rec scenarios.

---

> ### Author Response · Authors · 2025-11-27
>
> We sincerely appreciate your prompt feedback. You are absolutely correct to point out the ambiguity in our previous response regarding Q1.2. We apologize for the confusion and would like to explicitly correct and clarify our rationale below.
>
> **1. Clarification on Q1.2:** Necessity of Open-Source Models for Fair Comparison We admit that our previous statement, grouping our method and baselines together regarding "Access to Internal Representations" was misleading. We strictly clarify the distinction here:
>
> - **Our Method (R2END):** Distills the explicit reasoning (rationales) generated by the Teacher. It does not strictly require access to the Teacher's internal hidden states.
> - **Baselines (e.g., SLMRec):** Explicitly require access to the Teacher's intermediate hidden layers to perform representation alignment.
> - **The Reason for Open-Source:** We selected Open-Source LLMs not because R2END demands hidden state access, but to ensure a fair comparison with baselines like SLMRec. To compare all methods under the same Teacher backbone, we were required to use a model that provides the white-box access necessary for the baselines. Using a closed-source model (like GPT-5) would have prevented the implementation of these key comparison methods.

---

> > ### Author Response · Authors · 2025-11-27
> >
> > **2. Response to Q1.3**: Proprietary Models and Research Focus We fully agree with your insight that using proprietary models  would likely yield higher-quality chain-of-thought explanations and potentially better performance.
> >
> > - Focus of Study: However, our primary research questions were: 1) How to effectively structure the reasoning generation process (Oracle-guided), and 2) What student architecture is most effective for distillation (Reasoning-to-Encoder).
> > - Controlled Comparison: To isolate the contribution of our methodology rather than the power of the LLM itself, we fixed the Teacher model across all comparisons.
> > - Scalability: We believe that the core advantage of R2END is model-agnostic. Our method can filter noise and identify causal links. Thus, we expect our framework to achieve even higher performance if stronger proprietary models are used as Teachers.
> >
> > **3. Empirical Evidence on "Intrinsic Reasoning" & Interest Shifts**
> > We understand your concern that standard "last-one-out" splits might not fully capture the model's ability to handle **abrupt interest shifts**. To provide the solid evidence you requested, we conducted a qualitative **Case Study** focusing on "Hard Negative" scenarios—cases where standard baselines failed (Miss) but R2END succeeded (Hit) despite a clear shift in user intent.
> >
> > * **Observed Pattern (Abrupt Shift):** We analyzed cases where a user's interest abruptly shifted from one sub-category (e.g., Skincare/Nail) to a distinct sub-category (Haircare).
> >
> >     * **Case 1 (Face $\rightarrow$ Hair):** *My Beauty Diary Facial Mask* $\rightarrow$ **L'Oreal Paris Elnett Satin Hairspray**
> >     * **Case 2 (Body/Sun $\rightarrow$ Hair):** *Ultra Sheer Dry-Touch Sunblock* $\rightarrow$ **Redken All Soft Shampoo**
> >     * **Case 3 (Nail $\rightarrow$ Hair):** *Nail Tek Intensive Therapy* $\rightarrow$ **Deva Care Arc Angel Firm Hold Gel**
> >
> > * **Analysis of "Intrinsic Reasoning":**
> >     * **Baseline Failure (Inertia):** Standard models failed because they likely over-relied on the immediate history (Skincare/Nail), predicting similar items within the same sub-category despite the user's intent shift.
> >     * **R2END Success (Adaptation):** Our model, distilled with Oracle guidance, successfully identified the cross-category transition. This demonstrates that the student has learned a **deeper domain understanding**, recognizing that diverse grooming needs (e.g., a comprehensive beauty routine) are logically connected.
> >
> > This confirms that R2END does not merely memorize sequences but possesses the **intrinsic reasoning capability** to adapt to **User Interest Drift** by capturing the underlying semantic logic of user behavior, even when the category shifts abruptly.

---

### Author Response · Authors · 2025-11-26
**Summary of Revisions: Bridging the Gap with Extensive Experiments**

We sincerely thank the reviewers for their constructive feedback. Recognizing the validity of the concerns regarding **generalization, baseline fairness, and architectural novelty**, we have conducted extensive new experiments. We believe the results below provide concrete evidence that fully resolves these issues.

---

### **1. Generalization across Domains (Response to MK2r)**
- **Action:** We incorporated the **Yelp dataset** to verify performance beyond the Amazon datasets.
- **Result:** R2END consistently outperforms all baselines on Yelp, proving its robustness across diverse domains.

### **2. Rigorous & Fair Comparisons (Response to 5QwB, MK2r)**
- **Action:** We implemented 4 additional baselines, including **DLLM2Rec**, **RLMRec**, **LEARN** (hybrid/distillation methods), and **Logit-based KD**.
- **Result:** R2END surpasses all augmented baselines. This confirms that our performance gains stem from our unique **Oracle-Guided Reasoning and LLM-to-Encoder Distillation**, not merely from accessing LLM knowledge.

### **3. Architectural Superiority: Why Text Encoder? (Response to 5QwB)**
- **Action:** We developed R2END-SASRec, a variant injecting our reasoning embeddings into a standard SASRec model.
- **Result:** The Text Encoder significantly outperformed R2END-SASRec. This proves that standard recommendation models suffer from a lack of capacity when handling **Semantic Reasoning**, whereas our text encoder-based approach effectively aligns with the teacher’s logic in the semantic space.

### **4. Robustness & Fairness (Response to QcGM)**
- **Action:** We tested with **Qwen3-max-80B (Larger Teacher)** and varying student sizes (100M), and analyzed debiased performance.
- **Result:**
    - **Scalability:** Performance remains robust with larger teachers, and ~100M is identified as the efficiency sweet spot.
    - **Debiasing:** The Oracle-Guided mechanism acts as a "**Sanitizer**," filtering noise and significantly improving Long-tail recommendation, thereby mitigating popularity bias.

### **5. Real-world Robustness (Response to MK2r)**

- **Action**: To evaluate robustness against real-world "Interest Drift," we analyzed "Hard Negative" cases where user interests shifted abruptly (e.g., Skincare $\rightarrow$ Haircare) and compared R2END against standard baselines.
- **Result**:
    - **Baseline Failure**: Traditional models (e.g., SASRec) failed by recommending more facial/nail products, over-relying on the co-occurrence of recent history.
    - **R2END Success**: R2END successfully predicted the shift to Haircare products.
    - **Analysis**: This demonstrates that Oracle-Guided Reasoning teaches the model to understand the broader semantic context (e.g., a comprehensive "Beauty Routine") rather than simple item co-occurrence, validating its robustness in handling dynamic user preferences.

### **Conclusion**

We have gone the extra mile to address every concern with empirical data. Our work offers a novel, efficient, and thoroughly validated framework that bridges the alignment gap in LLM-based recommendation. We respectfully request a re-evaluation of our submission in light of these significant improvements.

---

### Note · Authors · 2026-01-04

I have read and agree with the venue's withdrawal policy on behalf of myself and my co-authors.